# Determination of the Dissociation Constants (p*K*ₐ) of Eight Amines of Importance in Carbon Capture: Computational Chemistry Calculations, and Artificial Neural Network Models

**Venkata Sai Priyatham Varma Alluri, William (Hoang Chi Hieu) Nguyen**  **and Amr Henni** *

Clean Energy Technologies Research Institute (CETRi), Acid Gas Removal Laboratory (AGRL),
Faculty of Engineering and Applied Science, University of Regina, Regina, SK S4S 0A2, Canada;
priyatham.alluri609@gmail.com (V.S.P.V.A.); william_nguyen_85@hotmail.com (W.N.)
* Correspondence: amr.henni@uregina.ca; Tel.: +1-(306)-585-4960

**Abstract:** This work focuses on determining the dissociation constants (p*K*ₐ) of eight amines, namely, 3-(Diethylamino) propylamine, 1,3-Diaminopentane, 3-Butoxypropylamine, 2-(Methylamino) ethanol, Bis(2-methoxyethyl) amine, α-Methylbenzylamine, 2-Aminoheptane, and 3-Amino-1-phenylbutane, within temperatures ranging from 293.15 K to 323.15 K. The thermodynamic properties of the protonated reactions were regressed from the p*K*ₐ work. In addition, the protonated order of both 3-(Diethylamino) propylamine and 1,3-Diaminopentane were determined using computational chemistry methods owing to their unsymmetrical structures. In addition to the experimental methods, the dissociation constants at the standard temperature (298.15 K) were also estimated using group functional models (paper–pencil) and computational methods. The computational methods include COSMO-RS and computational chemistry calculations. An artificial neural network (ANN) method was employed to model the data by collecting and combining the experimental properties to estimate the missing p*K*ₐ values. Although the ANN models can provide acceptable results, they depend on the availability of the data. Instead of using the experimental properties, they were generated using software such as Aspen Plus or CosmothermX. The simulated ANN model can also provide very good fits to the experimental constant values.

**Keywords:** p*K*ₐ; amines; group contribution; ANN

## 1. Introduction

Carbon dioxide ($CO_2$) is mainly generated by burning fossil fuels for human daily activities such as power generation, heating demands, and from refineries, steel, and cement production plants. $CO_2$ is considered one of the greenhouse gases that cause global warming [1]. There are many $CO_2$ capture technologies, such as using liquid solvents or solid sorbents to separate $CO_2$ from the other gases. For the flue gases with low $CO_2$ partial pressures, the aqueous amine solutions have been used because the solutions can quickly react with $CO_2$ while physical solvents are not very suitable to capture $CO_2$ at low pressures. For the different amine types, the reaction mechanisms between the amines and $CO_2$ are also very different [2]. As a result, the study of the protonation between amines and $CO_2$ is necessary to understand the reaction mechanism as well as model gas solubility in the solution [3].

Chowdhury et al. [4] and Tomizaki et al. [5] studied and reported the dissociation constants (p*K*ₐ) for cyclic amines, while Perrin [6] reported the dissociation constants for many alkanolamines. Both Versteeg et al. [7] and Sharma [8] studied the relationships between the dissociation constants and the reaction rate between $CO_2$ and aqueous amine solutions. The p*K*ₐ values of 2-(Butylamino)ethanol, m-Xylylenediamine, 3-Picolylamine, Isopentylamine, and 4-(Aminoethyl)-piperidine were recently measured and reported by Kumar et al. [9], while Nguyen et al. [10] reported the experimental values and explained

the protonation mechanisms by computational chemistry calculation (DFT) for 1,4-Bis(3-aminipropyl) piperazine, 1,3-Bis(aminomethyl) cyclohexane, Tris(2-aminoethyl) amine, and 1-Amino-4-methyl piperazine. In addition, the constant values can be used to determine amines' chemical and biological behaviors [9–12].

Some methods used to determine the dissociation constant values for amines include ultraviolet spectrophotometry, potentiometric titration, magnetic resonance, and conductimetric titration [13]. Among these methods, potentiometric titration is widely used because of its simplicity and convenience within the pH range of 2.00 to 11.00 [13–15]. In this study, the potentiometric titration method was used to determine and dissociation constant of Monoethanolamine (MEA), 3-(Diethylamino) propylamine, 1,3-Diaminopentane, 3-Butoxypropylamine, 2-(Methylamino) ethanol, Bis(2-methoxyethyl) amine, $\alpha$-Methylbenzylamine, 2-Aminoheptane, and 3-Amino-1-phenylbutane for the temperature range between 298.15 K to 313.15 K with an increment of 5 K. The average atmospheric pressure during the experiments was 95.8 ($\pm$0.8) kPa. The purpose of using monoethanolamine (MEA) is for validating the equipment. The compounds' structures, suppliers, and purities are listed in Table 1.

Based on the structures in Table 1, the studied amines included two diamines (DEAPA and EP) and six monoamines (BPA, MAE, BMOA, PEA, AH, and APB). All six monoamines were primary or secondary amines that have high kinetic and $CO_2$ uptakes. The diamines were primary and tertiary amines. In addition, instead of directly measuring the kinetic rates between the aqueous amine solutions with $CO_2$, the dissociation constants ($pK_a$) can be used to rank the reaction rates for the studied amines. Furthermore, the dissociation constant can be used for further modeling $CO_2$ absorptions into the aqueous solutions. The studied amines are mainly monoamines and tertiary amines; therefore, they are expected to have lower heat of absorption than polyamines. As a result, these amines have a high potential for $CO_2$ capture.

**Table 1.** Chemical name, structures, suppliers, and mass purity (%) of the studied amines.

| Full Name | IUPAC Name | Abbreviation Name | CAS Reg. No. | Molecular Structure | Molecular Mass (g·gmol$^{-1}$) | Supplier | Mass Fraction Purity (%) |
|---|---|---|---|---|---|---|---|
| Monoethanolamine | 2-Aminoethanol | MEA | 141-43-5 | | 61.08 | Sigma-Aldrich (Oakville, ON, Canada) | ≥99 |
| 3-(Diethylamino) propylamine | N′,N′-diethylpropane-1,3-diamine | DEAPA | 104-78-9 | | 130.23 | Sigma-Aldrich | ≥99 |
| 1,3-Diaminopentane | Pentane-1,3-diamine | EP | 589-37-7 | | 102.18 | Sigma-Aldrich | 98 |
| 3-Butoxypropylamine | 3-Butoxypropan-1-amine | BPA | 16499-88-0 | | 131.22 | Sigma-Aldrich | 99 |
| 2-(Methylamino) ethanol | 2-(Methylamino)ethanol | MAE | 109-83-1 | | 75.11 | Sigma-Aldrich | ≥98 |
| Bis(2-methoxyethyl) amine | 2-Methoxy-*N*-(2-methoxyethyl)ethanamine | BMOA | 11-95-5 | | 133.19 | Sigma-Aldrich | 98 |
| α-Methylbenzylamine | 1-Phenylethanamine | PEA | 618-36-0 | | 121.18 | Sigma-Aldrich | 99 |
| 2-Aminoheptane | [(2R)-heptan-2-yl]azanium | AH | 123-82-0 | | 115.22 | Sigma-Aldrich | 99 |
| 3-Amino-1-phenylbutane | 4-Phenylbutan-2-amine | APB | 22374-89-6 | | 149.24 | TCI (Portland, OR, USA) | 98 |

## 2. Chemicals, Apparatus, and Experimental Procedures

The nine amines, namely, Monoethanolamine (MEA) (≥99% in mass purity), 3-(Diethylamino) propylamine (DEAPA) (≥99% in mass purity), 1,3-Diaminopentane (EP) (98% in mass purity), 3-Butoxypropylamine (BPA) (99% in mass purity), 2-(Methylamino) ethanol (MAE) (≥98% in mass purity), Bis(2-methoxyethyl) amine (BMOA) (98% in mass purity), α-Methylbenzylamine (PEA) (99% in mass purity), 2-Aminoheptane (AH) (99% in mass purity), and 3-Amino-1-phenylbutane (APB) (98% in mass purity) were purchased from Sigma Aldrich.

To measure the pH of the aqueous amine solutions, a 270 Denver Instrument pH meter (Denver Instruments, Arvada, CO, USA) was used in the titration method. Three pH buffer solutions were purchased from VWR (Mississauga, ON, Canada) and used to calibrate the pH meter. Buffer solutions of 4.00, 7.00, and 10.00 were used, and the uncertainties supplied by the manufacturer (VWR) were 0.02, 0.00, and 0.01 for the 10.00, 7.00, and 4.00 buffer solutions, respectively. Table S1 in the Supporting Information (SI) section summarizes the calibration data for the pH meter.

The equipment and experimental procedures were validated by determining the $pK_a$ of monoethanolamine (MEA) and compared with the values in the literature [9–11,14,15]. Figure S1 in SI shows the comparisons of the dissociation constant values ($pK_a$) of MEA between this study and the data from the literature and is compared in Table S4. Hydrochloric acid solution 0.100 N (±0.002 N) was purchased from VWR International for the titration process. Nitrogen with 99.99% mass purity was purchased from Linde (Mississauga, ON, Canada) for flushing the space on the top of amine solutions to replace oxygen ($O_2$) and carbon dioxide ($CO_2$) present in the environment. The temperature of the solution inside the amine-storing beaker was controlled by an external water bath. The uncertainty of the temperature was 0.01 K. The aqueous amines were mixed to have a concentration of 0.0100 M within ±0.0002 M. The solutions were stirred at least for 20 min before starting the experiments.

In the beginning, 50 mL of the mixing solution was transferred into the storing beaker which had a stirring magnetic bar. During the experiment, the stirring speed of the bar was maintained at 30 rpm. The experiment started when the solution reached equilibrium and the temperature and pH became stable. The initial pH values of the solutions at equilibrium conditions were recorded. The storing beaker was always closed during the experiment to avoid oxidative degradation or vaporization. At each step of the titration process, 0.5 mL of the acid was added to the aqueous solution while stirring with the magnetic bar. After 20 to 30 s, the pH values were recorded as the mixtures reached equilibrium.

Measured pH and the details of the calculated $pK_a$ values for MEA, before the thermodynamic corrections, are presented in Table S2. Table S3 shows the values of $pK_a$ before and after the correction for MEA at 298.15 K.

## 3. Calculation of Dissociation Constant ($pK_a$)

To determine the dissociation constant values for the studied amines, the titration method was used to measure the pH of the solutions at each step after adding the acid to the solutions. The procedures employed are reported below to estimate the $pK_a$ values for MEA as well as the other amines. The protonated mechanism of MEA and water is represented in Equation (1). In Equation (2), {MEAH$^+$} is the protonated MEA molecule and {MEA} is the monoethanolamine molecule. Equation (2) was used to determine the $pK_a$ values before the thermodynamic correction [10],

$$MEAH^+ + H_2O \overset{K_a}{\leftrightarrow} MEA + H_3O^+ \tag{1}$$

$$pK_a{}^M = pH + \log \frac{\{MEAH^+\}}{\{MEA\}} \tag{2}$$

Equation (3) was used to calculate ionic strength (I), which is a function of the molecular concentration of ionization species ($C_i$) in the solution and the valency of species ($z_i$). The extended Debye–Hückel activity coefficient ($\gamma$) was calculated using Equation (4) with the adjustable parameter $k_i$. The value of the parameter was dependent on the ion sizes. The Supporting Information (SI) provides the details of the p$K_a$ calculations as well as thermodynamic corrections with the parameters of A and B from the literature [16] and the values of $k_i$ from Sumon et al. [17] Equation (5) was used to determine the true concentration of protonated MEA. For the first dissociation constant calculation, the value of I can be simply used as in Equation (3); however, the Albert et al. [13] method was used to determine I for the second p$K_a$, as reported in the literature [10]. Equation (6) was used to determine the dissociation constant after the thermodynamic correction (TC).

$$I = 0.5\sum C_i z_i^2 \tag{3}$$

$$\gamma_i = 10^{-\frac{A z_i^2 \sqrt{I}}{1 + B k_i \sqrt{I}}} \tag{4}$$

$$[MEAH^+] = \frac{\{MEAH^+\}}{\gamma_i} \tag{5}$$

$$pK_a = pK_a{}^M - TC \tag{6}$$

## 4. Results and Discussion

Based on the calculation steps mentioned above and in the Supporting Information section, the results of the first dissociation constants of MEA and the eight studied amines are listed in Table S4. The first p$K_a$ values of MEA for various temperatures are also plotted in Figure S1 and reported in the SI section to compare them with the literature data. All the first dissociation constants of the studied amines and MEA are plotted in Figure 1.

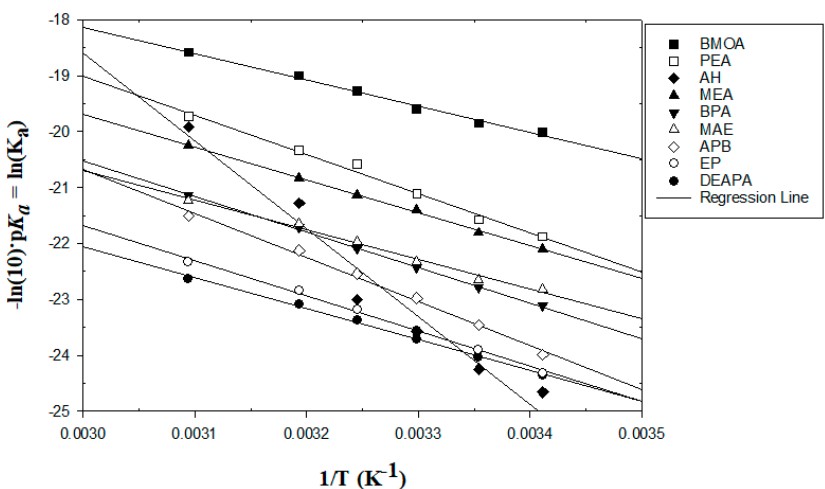

**Figure 1.** Linear relationship between the first dissociation constants and inverse temperatures of the amines studied.

First, the more negative values in Figure 1 have higher dissociation constant values. As the obtained data are reported in Table S4 and Figure 1, the six studied amines, namely, DEAPA, EP, APB, MAE, AH, and BPA, have higher dissociation constant values than MEA within the studied temperature range. Furthermore, two studied amines, BMOA and PEA, have lower constant values for the studied temperature range. In addition, AH has a high slope of the linear regression line which implies that the kinetic of the amine strongly

depended on the temperature and the concentration. The six amines with higher $pK_a$ values than MEA were expected to have higher reaction rates with $CO_2$ than MEA.

In addition, among the eight studied amines, two were diamines, DEAPA and EP. The values of the second dissociation constant are reported in Table S5 and Figure 2. As shown, the second $pK_a$ of DEAPA was higher than that of EP. Based on the molecular structures of the two amines, DEAPA included one primary and one tertiary amino group, while EP had two primary amino groups. The tertiary amine tended to react slower with $CO_2$ than the primary amine; however, DEAPA had a higher second $pK_a$ than EP, with two primary amino groups in the structure. Furthermore, tertiary amines have lower heat of absorption than primary amines. Therefore, DEAPA has great potential as an amine for $CO_2$ capture.

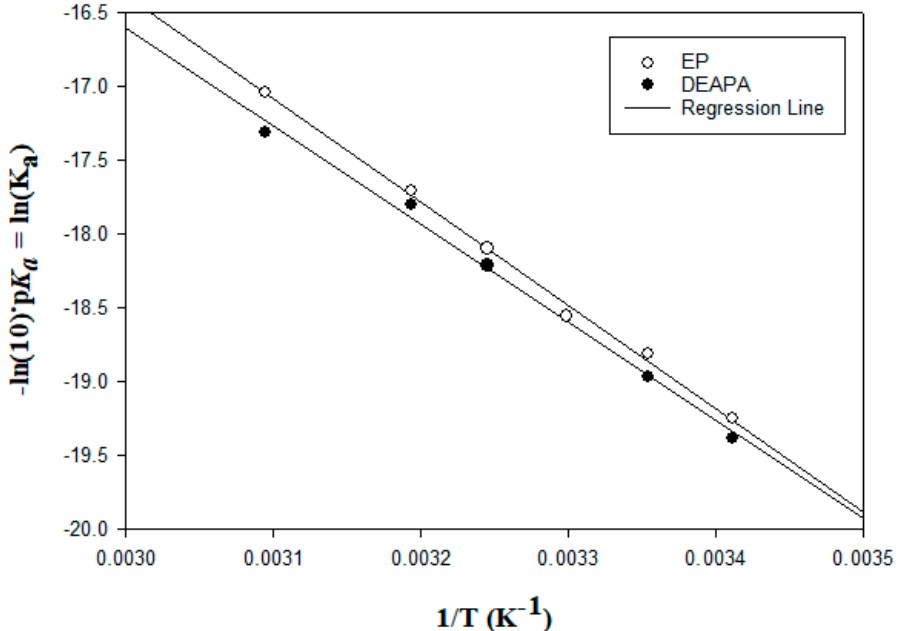

**Figure 2.** Linear relationship between the second dissociation constants with the inverse of temperature for diamines.

The coefficients of determination ($R_2$) of the linear regressions for Figures 1 and 2 are reported in Table S6.

The standard state enthalpy change ($\Delta H^0$, kJ·mol$^{-1}$) and entropy change ($\Delta S^0$, kJ·mol$^{-1}$·K$^{-1}$) can be determined using the van't Hoff equation [9,10,15]. These thermodynamic quantities ($\Delta H^0$ and $\Delta S^0$) can be regressed using data in Tables S4 and S5 and Figures 1 and 2 by applying Equation (7). In the equation, the slope and intercept of the linear regressions are ($\frac{-\Delta H^0}{R}$) and ($\frac{\Delta S^0}{R}$), respectively. The universal gas constant (R, kJ·mol$^{-1}$·K$^{-1}$) value is $8.3145 \times 10^{-3}$ kJ·mol$^{-1}$·K$^{-1}$. In addition, the standard state free energy of the reaction ($\Delta G^0$, kJ·mol$^{-1}$) was calculated using Equation (8). The calculated values of these thermodynamic quantities are reported in Tables 2–5.

$$\ln K_a = - \ln (10^{pK_a}) = \frac{\Delta S^0}{R} + \frac{-\Delta H^0}{RT} \qquad (7)$$

$$\Delta G^0 = RT \ln (10^{pK_a}) \qquad (8)$$

**Table 2.** The standard state enthalpy change ($\Delta H^0$/ kJ·mol$^{-1}$) and entropy change ($\Delta S^0$/ kJ·mol$^{-1}$·K$^{-1}$) of the first dissociation constants of the studied amines.

| Amine | $\Delta H^0$ (kJ·mol$^{-1}$) | $\Delta S^0$ (kJ·mol$^{-1}$·K$^{-1}$) |
|---|---|---|
| 3-(Diethylamino) propylamine | 46.05 | −0.05 |
| 1,3-Diaminopentane | 52.40 | −0.02 |
| 3-Butoxypropylamine | 52.83 | −0.01 |
| 2-(methylamino) ethanol | 44.27 | −0.04 |
| Bis(2-methoxyethyl) amine | 39.21 | −0.03 |
| α-Methylbenzylamine | 58.30 | 0.02 |
| 2-Aminoheptane | 130.9 | 0.24 |
| 3-Amino-1-phenylbutane | 65.68 | 0.03 |

**Table 3.** Standard state free energy of reaction ($\Delta G^0$, kJ·mol$^{-1}$) of the first dissociation constants of the studied amines.

| Amine | Temperature T/K | | | | | |
|---|---|---|---|---|---|---|
| | 293.15 | 298.15 | 303.15 | 308.15 | 313.15 | 323.15 |
| 3-(Diethylamino) propylamine | 59.38 | 59.59 | 59.78 | 59.88 | 60.13 | 60.81 |
| 1,3-Diaminopentane | 59.27 | 59.25 | 59.37 | 59.41 | 59.47 | 60.01 |
| 3-Butoxypropylamine | 56.35 | 56.51 | 56.53 | 56.58 | 56.53 | 56.79 |
| 2-(Methylamino) ethanol | 55.62 | 56.17 | 56.30 | 56.28 | 56.35 | 57.04 |
| Bis(2-methoxyethyl) amine | 48.77 | 49.20 | 49.39 | 49.38 | 49.46 | 49.93 |
| α-Methylbenzylamine | 53.32 | 53.48 | 53.22 | 52.74 | 52.94 | 53.02 |
| 2-Aminoheptane | 60.11 | 60.11 | 59.43 | 58.94 | 55.40 | 53.51 |
| 3-Amino-1-phenylbutane | 58.48 | 58.16 | 57.92 | 57.76 | 57.61 | 57.78 |

**Table 4.** The standard state enthalpy change ($\Delta H^0$/ kJ·mol$^{-1}$) and entropy change ($\Delta S^0$/ kJ·mol$^{-1}$·K$^{-1}$) of the second dissociation constants of 3-(Diethylamino) propylamine and 1,3-Diaminopentane.

| Amine | $\Delta H^0$ (kJ·mol$^{-1}$) | $\Delta S^0$ (kJ·mol$^{-1}$·K$^{-1}$) |
|---|---|---|
| 3-(Diethylamino) propylamine | 55.33 | 0.03 |
| 1,3-Diaminopentane | 58.18 | 0.04 |

**Table 5.** Standard state free energy of reaction ($\Delta G^0$, kJ·mol$^{-1}$) of the second dissociation constants of 3-(Diethylamino) propylamine and 1,3-Diaminopentane.

| Amine | Temperature T/K | | | | | |
|---|---|---|---|---|---|---|
| | 293.15 | 298.15 | 303.15 | 308.15 | 313.15 | 323.15 |
| 3-(Diethylamino) propylamine | 47.26 | 47.03 | 46.78 | 46.66 | 46.34 | 46.52 |
| 1,3-Diaminopentane | 1.35 | 1.29 | 1.24 | 1.19 | 1.14 | 1.07 |

## 5. Estimation Protonated Order for the Diamines

There are two diamines in this study, namely, 3-(Diethylamino) propylamine and 1,3-Diaminopentane. As mentioned above, 3-(Diethylamino) propylamine's structure has one primary and one tertiary amino group, while 1,3-Diaminopentane includes two unsymmetrical primary amino groups. It is important to know that the protonation order is required to model the solubility of CO$_2$ in amine solutions. To perform the study, commercial computational chemistry software (Gaussian (09 Revision B.01 SMP)) was used to optimize the molecular structures of the neutral amines with different models and basis sets. The molecular structures of the amines were built by using GaussView version 5.0.9. The optimization model was performed with implicit (without explicit water molecule)

solvent and polarizable continuum model (PCM) as default. Then, the optimization structures would be reoptimized with protons at different amino groups. The energy (E) of the protonated amines was used to suggest the preferred amino group for the first and second p$K_a$. The structures with lower energies suggested the highest stable structures. For the computational chemistry methods, Hartree–Fock (HF), density functional theory (DFT/B3LYP), and the Moller–Plesset perturbation second order (MP2) were used. The basis sets chosen were 6-311G+(d,p), B3LYP/3-21G, and B3LYP/6-311G+(d,p), or HF, DFT, and DFT, respectively. For the MP2 calculation, the chosen basis set was 6-311G+(d,p). For the energy notation in Equation (9), $E_1$ represented the potential energy surface (PES) when the proton was attached to the primary amino group on the carbon chain, while $E_2$ represented the PES when the proton was attached to the primary amino group on the branch for (1,3-Diaminopentane) or the tertiary amino group for 3-(Diethylamino propylamine). The two geometrical structures for the two amines were optimized with water as a solvent and a polarizable continuum model (PCM).

$$\Delta E \ (kJ \cdot mol^{-1}) = E_1 - E_2 \tag{9}$$

Based on the results reported in Table 6, the differences in PES ($\Delta E / kJ \cdot mol^{-1}$) were positive for all calculation methods for 3-(Diethylamino) propylamine. This means that the $E_1$ was higher than $E_2$ or the protonated tertiary amino group was more stable than the protonated primary amino group in the structure. The results do not agree with what is reported in the literature [9,10,16], which suggested that the primary amino group should be protonated before the tertiary amino group. However, the functional groups in the structure can strongly affect the order of protonation for the amines. This conclusion will be studied further in the later parts of this study. As for 1,3-Diaminopentane, the differences were all negative; this implies that the protonated amino group on the chain structure was more stable than the protonated amino group on the branch.

**Table 6.** The difference in potential energy surface (PES) between protonated 3-(Diethylamino) propylamine and 1,3-Diaminopentane structures.

| Model/Basis Set | $\Delta E$ (kJ·mol$^{-1}$) | |
| :---: | :---: | :---: |
| | **3-(Diethylamino) propylamine** | **1,3-Diaminopentane** |
| HF/6-311G+(d,p) | 64.91 | −7.13 |
| DFT/B3LYP/3-21G | 5.72 | −17.71 |
| DFT/B3LYP/6-311G+(d,p) | 14.93 | −6.13 |
| MP2/6-311G+(d,p) | 21.00 | −8.65 |

## 6. Computational Estimation for the Dissociation Constants

In addition to the experimental measurements, the dissociation constants could be estimated by other methods such as the PDS [16] (Perrin, Dempsey, and Serjeant) method, the new PDS [17] method, the QSSG [18] method, the COSMO-RS method, the Van't Hoff equation, the SHE [17] method, and the KHE [19] method. The first three methods were simply called functional group methods because the method suggested the base values for the primary, secondary, and tertiary groups. The functional groups around these amino groups would shift the base values. The shifted values depended on the type of functional groups and their positions. For the other methods, computational chemistry calculations were required to optimize the structure from which the Gibbs free energy would be determined before and after the protonation occurred. Based on the difference in the free energy, the values of dissociation constants can be determined.

### 6.1. Functional Group Estimations

The method can be used to estimate the first dissociation constant for the studied amines at 298.15 K based on the chemical structures. As mentioned above, the base values and the p$K_a$ shift were predetermined by optimizing the structures and the experimental

values of many chemicals; as a result, the estimated values' accuracies improved when more experimental values were available for optimization. For further improvements, Sumon et al. [17] and Qian et al. [18] included more data to reoptimize the base and shift values. Sumon's method is called the new PDS while Qian's method is called the QSSG method in this study. The suggested base and shift values for the three methods are summarized in Table S7. By applying these methods to determine the dissociation constant values for the studied amines, the first $pK_a$ at 298.15 K values are summarized in Table 7 below. 2-(Methylamino) ethanol was chosen to demonstrate the calculation method. Based on the structure in Table 1, this amine has a secondary amino group, $(CH_3)$– on the secondary amino, and –(OH) on the gamma ($\gamma$) position. For the PDS method, the first $pK_a$ value of the amine would be $11.15 - 0.2 - 1.1 \times 0.4 = 10.51$. For the new PDS method, the first $pK_a$ value would be 10.50, while the QSSR method suggests 10.2. By comparing the estimated values with the experimental values (9.84), the lowest error was 0.36 while the highest error was 0.67. Table 7 summarizes the estimated $pK_a$ values for the three methods. As mentioned above, 3-(Diethylamino) propylamine and 1,3-Diaminopentane could be protonated at different locations; therefore, the superscript V1 was used to represent the primary amino protonating on the main chain while V2 stood for the tertiary amino or the branch primary amino protonating, as shown in Figure 3.

**Table 7.** The estimated $pK_a$ values of the eight amines with three different methods at T = 298.15 K.

| Solvent | $pK_a$, PDS [19] | $pK_a$, Sumon [20] | $pK_a$, QSSR [21] | $pK_a$ [Exp.] |
|---|---|---|---|---|
| 3-(Diethylamino) propylamine[V1] | 10.63 | 10.44 | 10.24 | 10.44 |
| 3-(Diethylamino) propylamine[V2] | 10.37 | 10.46 | 10.28 | |
| 1,3-Diaminopentane[V1] | 10.64 | 10.46 | 10.28 | 10.38 |
| 1,3-Diaminopentane[V2] | 10.64 | 10.46 | 10.28 | |
| 3-Butoxypropylamine | 10.58 | 10.38 | 10.13 | 9.90 |
| 2-(Methylamino) ethanol | 10.51 | 10.50 | 10.20 | 9.84 |
| Bis(2-methoxyethyl) amine | 10.19 | 9.98 | 9.24 | 8.62 |
| $\alpha$-Methylbenzylamine | 10.77 | 10.60 | 10.60 | 9.37 |
| 2-Aminoheptane | 10.77 | 10.60 | 10.60 | 10.53 |
| 3-Amino-1-phenylbutane | 10.77 | 10.60 | 10.60 | 10.19 |

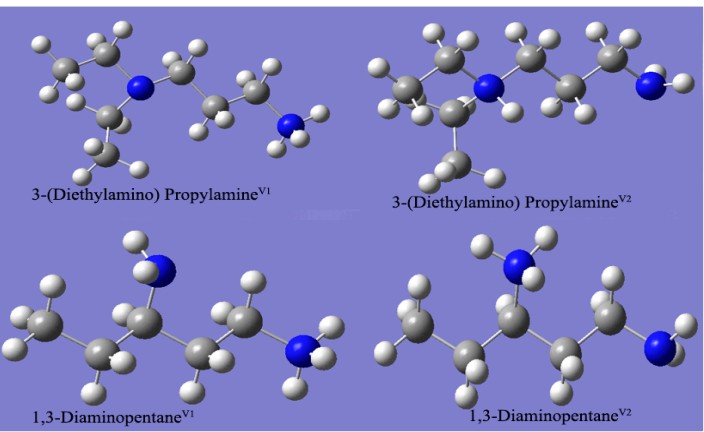

**Figure 3.** The protonating possibilities for 3-(Diethylamino) propylamine and 1,3-Diaminopentane. Blue: nitrogen, grey: carbon, and white: hydrogen.

*6.2. Computational Chemistry Estimation*

In addition to the functional group estimations such as the PDS or QSSR methods, the dissociation constants values of the amines at the standard conditions can be estimated by computational chemistry calculations. In particular, the SHE [17] method applied

quantum computational chemistry calculations to determine amine p$K_a$ values. In general, the dissociation mechanism of an amine was reported in Equation (10). By combing Equations (8) and (10), the dissociation constant at standard condition was determined by Equation (11) with R = 1.986 × 10$^{-3}$ kcal·K$^{-1}$·mol$^{-1}$, and the unit of energy is kcal·mol$^{-1}$. Equation (12) was derived by expanding and simplifying Equation (11) as reported in the literature [17]:

$$BH^+_{(aq)} \leftrightarrow B_{(aq)} + H^+_{(aq)} \tag{10}$$

$$pK_a = \frac{\left[ G_{(aq)}\left(H^+\right) + G_{(aq)}(B) - G_{(aq)}\left(BH^+\right) \right]}{1.3643} \tag{11}$$

$$pK_a = \frac{\left[ G_{(aq)}\left(H^+\right) + \Delta_h E_{el} + \Delta_h \Delta G_{non-el} + \Delta_h E_{nuc} \right]}{1.3643} \tag{12}$$

The values of $G_{(aq)}\left(H^+\right)$ = −270.3 kcal·mol$^{-1}$, $\Delta_h E_{nuc}$ = −9.4 kcal·mol$^{-1}$, and $\Delta_h \Delta G_{non-el}$ = 0 kcal·mol$^{-1}$ are reported in the literature [17]. Values of $\Delta_h E_{el}$ were determined by optimization of the molecular structures. In addition, the SHE method used the implicit–explicit model, which added one additional water molecule to the amine. In addition, an empirical correction factor was included to minimize the errors between the estimated experimental values as in Equation (13). The values of C were −1.7 for cyclic amines and −0.7 for acyclic amines [17].

$$pK_a = \frac{\left[ -270.3 + E_{el}(B \cdot H_2O) - E_{el}\left(BH^+ \cdot H_2O\right) - 9.4 \right]}{1.3643} + C \tag{13}$$

However, as in Equation (11), the dissociation constant values could be calculated since $G_{(aq)}(B)$ and $G_{(aq)}\left(BH^+\right)$ could be found directly without further simplification as in the SHE method. This method is called the theoretical method in this study. All values of $G_{(aq)}(B)$, $G_{(aq)}\left(BH^+\right)$, $E_{el}(B \cdot H_2O)$, and $E_{el}\left(BH^+ \cdot H_2O\right)$ were determined using Gaussian (09 Revision B.01 SMP). These molecular structures of the amines and amines–water were built using GaussView version 5.0.9. For the theoretical method, the structures were optimized using different methods (DFT/B3LYP and MP2) and the basis sets (6-31G(d) and 6-311G++(d,p)), while the SHE method used the DFT calculation method and 6-31G(d) basis set. All the amines' geometrical structures were optimized with water as a solvent and the polarizable continuum model (PCM).

In addition to the quantum computational chemistry calculation methods mentioned, COSMO-RS can be used to determine the dissociation constant values with the same concept as in Equation (11) at the standard temperature (T = 298.15 K). In this study, CosmothermX (Version C30_1201) software was used to estimate the p$K_a$ values for the studied amines with the TZVP calculation method. The results from these calculations were compared to the experimental values in Table 8. As mentioned above, the V1 superscript stands for the protonated main chain amino group while the V2 superscript stands for the branched or tertiary amino group.

By comparing the group functional methods in Table 7, the original PDS method tended to predict higher values than Sumon's method, which led to higher values than Qian's method. In general, the estimated values by Qian's method were closest to the experimental values than the other methods, on average. It should be mentioned that Qian's method had more functional groups than the original PDS and Sumon's method. In addition, Qian's method included the effects of the group functional group on the $\varepsilon$ (theta) position while the original PDS and Sumon's methods included only the $\delta$ (delta) position. Therefore, the estimated values by Qian's method were the closest to the experimental values.

**Table 8.** Comparison between the estimated p$K_a$ values by different methods with experimental dissociation constants.

| Amine | SHE Method | | Theoretical Method | | | Cosmo-RS Method | Experimental Values |
| --- | --- | --- | --- | --- | --- | --- | --- |
| | p$K_a$ without Correction | p$K_a$ with Correction | DFT/B3LYP 6-31G(d) | DFT/B3LYP 6-311G++(d,p) | MP2 6-31G(d) | | |
| 3-(Diethylamino) propylamine[V1] | **8.81** | 8.11 | 6.95 | 3.69 | 6.60 | 10.36 | 10.44 |
| 3-(Diethylamino) propylamine[V2] | 10.27 | 9.57 | 8.74 | 6.81 | 8.23 | 9.89 | |
| 1,3-Diaminopentane[V1] | 9.47 | 8.77 | 5.01 | 3.21 | 5.38 | 10.65 | 10.38 |
| 1,3-Diaminopentane[V2] | 8.42 | 7.72 | 7.99 | 4.11 | 7.16 | 9.39 | |
| 3-Butoxypropylamine | 8.18 | 7.48 | 7.88 | 4.94 | 6.24 | 10.02 | 9.90 |
| 2-(Methylamino) ethanol | 8.32 | 7.62 | 6.51 | 3.80 | 6.07 | 9.50 | 9.84 |
| Bis(2-methoxyethyl) amine | 6.65 | 5.95 | 4.45 | 1.70 | 3.76 | 7.07 | 8.62 |
| $\alpha$-Methylbenzylamine | 7.69 | 6.49 | 5.20 | 2.13 | 4.74 | 9.29 | 9.37 |
| 2-Aminoheptane | 8.78 | 8.08 | 7.21 | 4.30 | 6.31 | 10.43 | 10.53 |
| 3-Amino-1-phenylbutane | 8.30 | 7.10 | 6.20 | 3.34 | 5.47 | 9.83 | 10.19 |

The value in bold was used as an example.

In addition, when comparing the values between the group functional methods and chemistry computational methods in Tables 7 and 8, the group functional methods were found to be more accurate than the computational methods. The functional group methods were also much faster and cheaper than the computational methods as they did not require high-performance computers and complex software while providing reasonable results. In addition, the estimates obtained using the Cosmo-RS calculation using CosmothermX (version C30_1201) were much better than both the SHE and the classical methods.

As mentioned earlier, 3-(Diethylamino) propylamine's first $pK_a$ can be associated with the tertiary amino position or V2 conformation in Figure 3. By comparing the estimated values by group functional methods and chemistry computational method (except the Cosmo-RS method), the $pK_a$ values for the tertiary protonated scenario were closer to the experimental value than the first protonated scenario. On the contrary, the group functional methods estimated the same values for 1,3-Diaminopentane for both cases. The computational methods with different basis sets have different values for dissociation constants for the different amine geometries. In particular, the SHE method and Cosmo-RS calculation provided the closest values to the experimental values while the classical methods estimated the best values, although all the values did not perfectly match the experimental results. According to the structure presented in Table 1, the chain is considered the longest structure while the branch is attached to the chain. Logically, one primary amino belongs to the chain while the other is the branch. By the group contributions, the effect of the primary on the chain and branch is the same for each group. Although the ethyl group effect on the $pK_a$ values was not studied, it will shift the original values. As observed in Table S7, the methyl group will reduce the original values; similarly, the ethyl group would have similar behavior with different values. By combining the logical analysis with the results reported in Table 6, the primary amino group attached to the chain would probably be protonated before the branched amino group.

## 7. Estimation of the Dissociation Constants at Various Temperatures by Computational Chemistry

The above methods were used to predict the $pK_a$ values for the amines at the standard temperature (T = 298.15 K). However, it was necessary to estimate the constants at different temperatures. Therefore, to determine the $pK_a$ values at various temperatures, a thermodynamic cycle would be employed. In this study, two different cycles would be used to calculate the constant values. Figure 4 represents a thermodynamic cycle which is called the "hydronium cycle". Equation (14) was derived from Equation (11), as reported in the literature [20], where $\Delta G_s^*(BH^+)$, $\Delta G_s^*(H_2O)$, $\Delta G_s^*(B)$, and $\Delta G_s^*(H_3O^+)$ are the free solvation energies of $BH^+$, $H_2O$, B, and $H_3O^+$, respectively. The gas phase acidity of $BH^+$ is represented by $\Delta G_{gas}^*$ and is defined by Equation (15) [20], while $G_{gas}^0(B)$, $G_{gas}^0(H_3O^+)$, $G_{gas}^0(H_2O)$, and $G_{gas}^0(BH^+)$ are the Gibbs free energies for B, $H_3O^+$, $H_2O$, and $BH^+$ in the gas phase. In addition, $\Delta G^{0*}$ is the free energy change which is associated with moving from the standard state that uses a concentration of 1 atm in the gas phase and 1 mol·L$^{-1}$ in the aqueous phase to a standard state that uses a concentration of 1·mol·L$^{-1}$ in both the gaseous and aqueous phases. Equation (16) defines the values of $\Delta G^{0*}$ for different temperatures [20].

$$pK_a = \frac{\left[\Delta G_{gas}^0 + \Delta G_s^*(B) - \Delta G_s^*(BH^+) + \Delta G_s^*(H_3O^+) - \Delta G_s^*(H_2O) + \Delta G^{0*}\right]}{RT\ln(10)} \quad (14)$$

$$\Delta G_{gas}^0 = G_{gas}^0(B) + G_{gas}^0(H_3O^+) - G_{gas}^0(BH^+) - G_{gas}^0(H_2O) \quad (15)$$

$$\Delta G^{0*} = RT\ln(24.46) \quad (16)$$

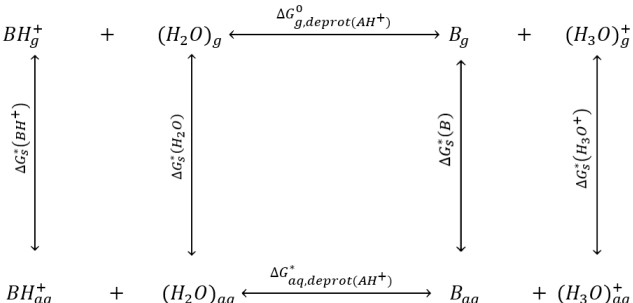

**Figure 4.** Hydronium thermodynamic cycle for calculating the dissociation constant.

The Gibbs free energy of the amines in the gaseous phase at various temperatures can be calculated by computational chemistry calculation. The solvation energies at various temperatures can be determined by Equation (17), as in the literature [20]. In the equation, $CavE_{298}$ and $\Delta G_s(298)$ are the cavitation energy and free energy of solvation at 298 K, respectively, which can be calculated by computational chemistry. The computational chemistry calculations can be used to determine the solvation-free energy of water ($H_2O$) and hydronium ($H_3O^+$) at standard temperature (298.15 K). The values of $-4.64$, $-5.05$, and $-5.12$ kcal·mol$^{-1}$ of water for three calculations (DFT/B3LYP/6-31G(d), DFT/B3LYP/6-311++G(d,p), and MP2/6-31G(d), respectively) at 298.15 K are very different from the experimental value of $-6.32$ kcal·mol$^{-1}$ reported in the literature [21]. Similarly, the values of $-77.63$, $-77.42$, and $-78.45$ kcal·mol$^{-1}$ of hydronium ($H_3O^+$) for three calculations (DFT/B3LYP/6-31G(d), DFT/B3LYP/6-311++G(d,p), and MP2/6-31G(d), respectively) at 298.15 K are much lower than the experimental values of $-110.3$ kcal·mol$^{-1}$ found in the literature [21]. Therefore, this study employed the experimental values of $-6.32$ kcal·mol$^{-1}$ and $-110.3$ kcal·mol$^{-1}$ at 298.15 K to improve the accuracy. Similar to the previous calculations, the computational chemistry calculations were performed by the methods and basis sets as DFT/B3LYP/6-31G(d), DFT/B3LYP/6-311++G(d,p), and MP2/6-31G(d), while the solvation model was the polarizable continuum model (PCM) with water as the solvent. The results of the calculations are summarized in Table 9.

$$\Delta G_s(T) = \Delta G_s(298) + (T - 298)\frac{CavE_{298}}{298} \tag{17}$$

By comparing the calculation values in Table 9 and the experimental values in Table S4, at the various temperatures, the calculation values obtained were very far from the experimental values. The differences could be the result of the errors between calculations and experimental values for each calculation or species in the cycle. For example, the significant differences for solvation-free energy of water and $H_3O^+$ between computational chemistry calculations and experimental values were observed previously. Therefore, more species or components involved in the thermodynamic cycle will result in more errors. By comparison, 3-(Diethylamino) propylamine with the protonated tertiary amino had constant values which were closer to the experimental values. Similarly, 1,3-Diaminopentane with the protonated chain primary amino had constant values close to the experimental results. Therefore, the calculations confirm the reaction order or mechanism of the two diamines as discussed previously. Furthermore, to reduce the components in the cycle in Figure 4, the hydronium $H_3O^+$ was replaced by the proton ($H^+$) while the water molecule was completely removed, as in the protonated thermodynamic cycle in Figure 5.

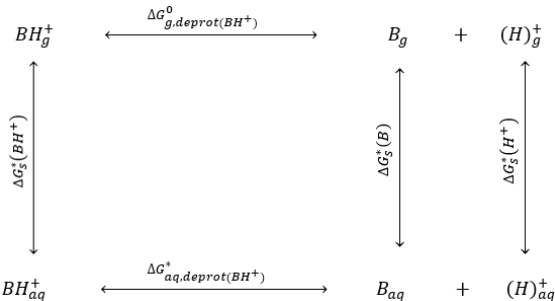

**Figure 5.** Protonated thermodynamic cycle for calculating the dissociation constants.

Because the proton does not include any electron, it could not be used in computational chemistry calculation optimization. The gas phase free energies of protons at various temperatures were obtained by fitting the data in the literature [22] from 0 K to 400 K. The values are reported in Table S8 in the SI section. In addition, Equations (18) and (19) were applied for the new thermodynamic cycle. In the equations, $\Delta G_s^*(B)$, $\Delta G_s^*(BH^+)$, and $\Delta G_s^*(H^+)$ were the free solvation energies of B, $BH^+$, and $H^+$, while $G_{gas}^0(B)$, $G_{gas}^0(H^+)$, and $G_{gas}^0(BH^+)$ were Gibbs free energies of B, $H^+$, and $BH^+$ in the gas phase. The calculations were performed the same as previously, while the cavitation energy of the proton was calculated with DTF/B3LYP/6-311G(d,p). The calculation results are summarized in Table 10.

$$pK_a = \frac{\left[\Delta G_{gas}^0 + \Delta G_s^*(B) - \Delta G_s^*(BH^+) + \Delta G_s^*(H^+) + \Delta G^{0*}\right]}{RT\ln(10)} \quad (18)$$

$$\Delta G_{gas}^0 = G_{gas}^0(B) + G_{gas}^0(H^+) - G_{gas}^0(BH^+) \quad (19)$$

By comparing the calculation results in Tables 9 and 10 and the experimental measurements in Table S4 for the first dissociation constants ($pK_a$), although the calculation results which were based on the protonated thermodynamic cycle were not very close to the experimental measurements, the protonated thermodynamic cycle calculations were much better than the hydronium thermodynamic cycle calculation results. The improvements could be the result of using the experimental data of the proton, rather than using all the calculation results of the proton. In addition, the basis set of 6-311G++(d,p) underestimated the calculation results while the basis set of 6-31G(d) overestimated the results. In addition, 3-(Diethylamino) propylamine with the protonated tertiary amino (V2) had constant values closer to the experimental values than the protonated primary amino version (V1), on average. For 1,3-Diaminopentane, the protonated chain primary amino 1,3-Diaminopentane (V1) had dissociation constant values closer to the experimental results than the protonated branch primary amino 1,3-Diaminopentane (V2) on average. The calculations confirm the reaction order or mechanism of the two diamines which have been discussed previously.

**Table 9.** Dissociation constant values of the studied amines at various temperatures (K) for DFT1 (DFT/B3LYP/6-31G(d)), DFT2 (DFT/B3LYP/6-311G++(d,p)), and MP2 (MP2/6-31G(d)) calculations by hydronium thermodynamic cycle.

| Amine | Temperature/K | | | | | | | | | | | | | | | | | |
|---|---|---|---|---|---|---|---|---|---|---|---|---|---|---|---|---|---|---|
| | 293.15 | | | 298.15 | | | 303.15 | | | 308.15 | | | 313.15 | | | 323.15 | | |
| | DFT1 | DFT2 | MP2 | DFT1 | DFT2 | MP2 | DFT1 | DFT2 | MP2 | DFT1 | DFT2 | MP2 | DFT1 | DFT2 | MP2 | DFT1 | DFT2 | MP2 |
| 3-(Diethylamino) propylamine[V1] | 7.34 | 7.43 | 7.87 | 7.24 | 7.32 | 7.77 | 7.14 | 7.23 | 7.66 | 7.05 | 7.13 | 7.56 | 6.96 | 7.04 | 7.47 | 6.79 | 6.87 | 7.28 |
| 3-(Diethylamino) propylamine[V2] | 9.16 | 10.61 | 9.53 | 9.03 | 10.45 | 9.40 | 8.91 | 10.31 | 9.27 | 8.79 | 10.16 | 9.15 | 8.68 | 10.02 | 9.03 | 8.46 | 9.76 | 8.80 |
| 1,3-Diaminopentane[V1] | 8.41 | 7.85 | 8.44 | 8.29 | 7.75 | 8.32 | 8.17 | 7.64 | 8.21 | 8.06 | 7.54 | 8.10 | 7.96 | 7.44 | 7.99 | 7.76 | 7.25 | 7.79 |
| 1,3-Diaminopentane[V2] | 5.34 | 6.94 | 6.62 | 5.30 | 6.94 | 6.63 | 5.26 | 6.77 | 6.46 | 5.23 | 6.69 | 6.38 | 5.19 | 6.61 | 6.30 | 5.12 | 6.45 | 6.16 |
| 3-Butoxypropylamine | 8.32 | 8.71 | 7.50 | 8.18 | 8.57 | 7.40 | 8.04 | 8.44 | 7.31 | 7.91 | 8.31 | 7.21 | 7.79 | 8.18 | 7.12 | 7.55 | 7.94 | 6.95 |
| 2-(Methylamino) ethanol | 6.90 | 7.54 | 7.33 | 6.81 | 7.43 | 7.24 | 6.72 | 7.33 | 7.14 | 6.63 | 7.24 | 7.05 | 6.55 | 7.15 | 6.96 | 6.40 | 6.97 | 6.79 |
| Bis(2-methoxyethyl) amine | 4.79 | 5.40 | 4.98 | 4.74 | 5.34 | 4.93 | 4.69 | 5.28 | 4.88 | 4.65 | 5.22 | 4.83 | 4.60 | 5.17 | 4.78 | 4.51 | 5.06 | 4.69 |
| α-Methylbenzylamine | 5.56 | 5.84 | 5.97 | 5.49 | 5.77 | 5.90 | 5.43 | 5.70 | 5.84 | 5.37 | 5.63 | 5.77 | 5.31 | 5.57 | 5.71 | 5.20 | 5.44 | 5.58 |
| 2-Aminoheptane | 7.60 | 8.04 | 7.58 | 7.50 | 7.93 | 7.48 | 7.41 | 7.83 | 7.38 | 7.32 | 7.73 | 7.29 | 7.23 | 7.64 | 7.20 | 7.06 | 7.45 | 7.02 |
| 3-Amino-1-phenylbutane | 6.58 | 7.07 | 6.72 | 6.49 | 6.98 | 6.63 | 6.41 | 6.88 | 6.55 | 6.33 | 6.80 | 6.47 | 6.26 | 6.71 | 6.39 | 6.11 | 6.55 | 6.24 |

**Table 10.** Dissociation constant values of the studied amines at various temperatures (K) for DFT1 (DFT/B3LYP/6-31G(d)), DFT2 (DFT/B3LYP/6-311G++(d,p)), and MP2 (MP2/6-31G(d)) calculations by protonated thermodynamic cycle.

| Amine | Temperature/K | | | | | | | | | | | | | | | | | |
|---|---|---|---|---|---|---|---|---|---|---|---|---|---|---|---|---|---|---|
| | 293.15 | | | 298.15 | | | 303.15 | | | 308.15 | | | 313.15 | | | 323.15 | | |
| | DFT1 | DFT2 | MP2 | DFT1 | DFT2 | MP2 | DFT1 | DFT2 | MP2 | DFT1 | DFT2 | MP2 | DFT1 | DFT2 | MP2 | DFT1 | DFT2 | MP2 |
| 3-(Diethylamino) propylamine[V1] | 10.45 | 7.13 | 10.09 | 10.19 | 6.93 | 9.85 | 9.95 | 6.74 | 9.61 | 9.71 | 6.56 | 9.38 | 9.48 | 6.38 | 9.16 | 9.04 | 6.03 | 8.73 |
| 3-(Diethylamino) propylamine[V2] | 12.27 | 10.31 | 11.75 | 11.99 | 10.06 | 11.48 | 11.72 | 9.82 | 11.22 | 11.45 | 9.59 | 10.96 | 11.20 | 9.36 | 10.72 | 10.71 | 8.93 | 10.24 |
| 1,3-Diaminopentane[V1] | 11.51 | 7.56 | 10.66 | 11.24 | 7.35 | 10.40 | 10.98 | 7.16 | 10.15 | 10.73 | 6.96 | 9.91 | 10.48 | 6.78 | 9.68 | 10.01 | 6.42 | 9.24 |
| 1,3-Diaminopentane[V2] | 8.45 | 6.64 | 8.84 | 8.26 | 6.46 | 8.62 | 8.07 | 6.28 | 8.41 | 7.89 | 6.11 | 8.20 | 7.71 | 5.94 | 7.99 | 7.37 | 5.62 | 7.61 |
| 3-Butoxypropylamine | 11.42 | 8.42 | 9.72 | 11.13 | 8.18 | 9.48 | 10.85 | 7.95 | 9.25 | 10.58 | 7.73 | 9.03 | 10.31 | 7.52 | 8.81 | 9.81 | 7.11 | 8.40 |
| 2-(Methylamino) ethanol | 10.00 | 7.24 | 9.55 | 9.76 | 7.04 | 9.32 | 9.52 | 6.85 | 9.09 | 9.29 | 6.66 | 8.87 | 9.07 | 6.48 | 8.65 | 8.65 | 6.14 | 8.24 |
| Bis(2-methoxyethyl) amine | 7.90 | 5.11 | 7.20 | 7.70 | 4.95 | 7.01 | 7.50 | 4.80 | 6.82 | 7.31 | 4.65 | 6.64 | 7.12 | 4.51 | 6.47 | 6.77 | 4.23 | 6.14 |
| α-Methylbenzylamine | 8.67 | 5.54 | 8.19 | 8.45 | 5.38 | 7.98 | 8.24 | 5.21 | 7.78 | 8.03 | 5.06 | 7.58 | 7.83 | 4.90 | 7.39 | 7.46 | 4.61 | 7.03 |
| 2-Aminoheptane | 10.71 | 7.75 | 9.80 | 10.46 | 7.54 | 9.56 | 10.22 | 7.35 | 9.33 | 9.98 | 7.16 | 9.10 | 9.75 | 6.97 | 8.89 | 9.32 | 6.62 | 8.47 |
| 3-Amino-1-phenylbutane | 9.69 | 6.77 | 8.94 | 9.45 | 6.58 | 8.71 | 9.22 | 6.40 | 8.49 | 9.00 | 6.22 | 8.28 | 8.78 | 6.05 | 8.08 | 8.37 | 5.72 | 7.68 |

### 8. The Artificial Neural Network (ANN) Models for Estimating the Constant Values of the Studied Amines

*8.1. ANN Model with Input Parameters*

As observed in the results and previous discussions, the dissociation constants of the studied amines at various temperatures were quite far from the experimental measurements. Although the measurements provided the most accurate values, the purchase of the chemicals is expensive in addition to the cost of waste disposal. In addition, chemists and chemical engineers have a great interest in estimating the constant values of chemicals even before their synthesis. Therefore, many researchers have attempted estimating the constant values at various temperatures with higher accuracies.

For computational chemistry estimation, Khalili et al. [12] used the commercial software Gaussian to predict the dissociation constant values of 17 amines at 298.15 K with $\pm 0.68$ $pK_a$ unit accuracy. Based on the Khalili–Henni–East (KHE) method [12], Sumon et al. [20] performed the calculations and improved the estimations to within $\pm 0.28$ $pK_a$ unit accuracy at the standard temperature. Although computational chemistry calculations can be applied to estimate the $pK_a$ for chemicals with known structures, the method could consume much CPU time and memory for large molecules. In addition to quantum chemistry calculations and quantitative structure–property relationship (QSPR) methods, artificial neural networks (ANNs) can be used to substitute experimental measurements. In summary, an ANN is a mathematical model which imitates the human brain to analyze information [23]. One of the advantages of an ANN is its flexibility to be applied in many fields using the nonlinear relationship in complicated systems without deep knowledge of the model; therefore, ANN applications have become very popular in solving engineering as well as scientific problems [24,25]. Habibi-Yangjeh et al. [25] predicted the dissociation constants of many different benzoic acids and phenol at standard temperature (298.15 K) by combining QSPR and ANN. The squared correlation coefficients ($R^2$) of the ANN model for training, validation, and prediction were 0.9926, 0.9943, and 0.9939, respectively. Many scientists have combined two models (ANN and QSPR) to estimate the constant values of chemicals. One of the challenges was to convert chemical structures to numerical information to feed into the ANN model. Another challenge was that these works were performed at the standard temperature (298.15 K). In this study, the ANN model is used for predicting $pK_a$ values for the amines at various temperatures.

First of all, data were collected to feed into the model (input data) and references (output) were obtained. The data can generally be divided into three segments, which included (1) properties that identified the chemicals (molecular weight, temperature, number of H, N, C, and O atoms), (2) properties associated with the dissociation constants (density, viscosity, refractive index, and sound velocity), and (3) referenced output (experimental measurement $pK_a$ values). In this study, 1140 data points of 31 sets of amines associated with $CO_2$ capture were collected and are listed in Table S9 in the SI section.

In the ANN study, the data were divided into three groups for training, validation, and testing. For training purposes, of the entire dataset, 70% of the data was randomly chosen. Similarly, of the entire dataset, 15% was randomly chosen for validation while the remaining 15% of the dataset was selected for testing the ANN model. In addition, this study optimized the number of hidden layers, the number of neurons in each hidden layer, and the number of epochs with the optimal condition of minimizing the mean squared error (MSE) in the ANN model. Initially, only one hidden layer was applied while the number of neurons varied from 30 to 80. Figure S2 shows the performances which included the correlation® and mean squared error (MSE) of the ANN model.

Based on Figure S2, the ANN model performed well with one hidden layer when the number of neurons was equal to 40. In particular, the $R_{overall} = 0.86156$, $MSE_{train} = 0.0288$, $MSE_{validation} = 0.0460$, and $MSE_{test} = 0.09044$. To study the effect of additional hidden layers, an extra hidden layer was added with 40 neurons for the first layer, while the number of neurons for the extra hidden layer was varied from 10 to 50. Figure S3 shows the performances with R and MSE of the ANN model. As observed in Figure S3, the

performances of the ANN model significantly improved by introducing the second hidden layer into the model. In particular, the model was optimized when the number of neurons was 39 with $R_{overall}$ = 0.9075, $MSE_{train}$ = 0.0069, $MSE_{validation}$ = 0.0372, and $MSE_{test}$ = 0.0597.

For further improvement, an additional third hidden layer was introduced into the ANN model. However, the performances of the ANN model did not improve; therefore, as a final scenario, this study used 40 and 39 neurons for the first and second hidden layers, respectively. The neurons, weights, and biases for the first, second, and output layers are reported in Tables 11–13, respectively. The overfitting in the ANN model happened when the model could predict the output with very high accuracy in the training process; however, the predictions would not fit well in the validating and testing processes. Therefore, balancing the accuracies between all processes is necessary to avoid overfitting issues. In addition, the studied ANN model chose the right balance between the variables and the hyperparameters which were the number of epochs, neurons, and batch sizes.

The ANN model architecture was described in Figure S4 for this study. The process of determining the best overall performance by running the model with many epochs varying from 140 to 200 is presented in Figure S5. Based on the results, the best performance for the overall MSE was at 166 epochs with a value of 0.026.

Figure S6 is a parity plot between the output and target for the entire dataset which included training, validation, and testing. The overall R was higher than 0.9; therefore, the ANN model provided a good estimation of the dissociation constants for a wide range of temperatures. Figure S7 shows the error histogram with only a few outliers. Based on the figure, the majority of the errors were close to the value of $-0.06303$, which almost overlaps with the zero line.

**Table 11.** Neurons, weights, and biases for the first hidden layer for the ANN model in this study.

| Neuron | Temperature | Molecular Weight | Number of C | Number of H | Number of N | Number of O | Refractive Index | Sound Velocity | Density | Viscosity | Bias |
|---|---|---|---|---|---|---|---|---|---|---|---|
| 1 | −0.1606 | 0.0418 | −0.2179 | 0.1791 | −0.5387 | 0.5581 | −0.0116 | 0.5601 | 0.3960 | 0.5408 | 0.1280 |
| 2 | −0.0019 | 0.2824 | −0.4485 | 0.6275 | −0.4088 | −0.0983 | 0.2342 | 0.2044 | −0.2735 | −0.9534 | 0.4230 |
| 3 | 0.3512 | 0.5736 | −0.4307 | −0.0813 | −0.7615 | −0.0966 | 0.6547 | 0.1445 | −0.7132 | −0.5333 | 0.3857 |
| 4 | −0.7657 | 0.4059 | 0.2351 | −0.1342 | 0.4731 | 0.5649 | −0.5287 | 0.4273 | −0.5889 | 0.4599 | 0.0970 |
| 5 | −0.5625 | 0.6416 | −0.0182 | −0.5174 | 0.3660 | 0.6449 | −0.1326 | −0.6361 | −0.2539 | 0.2892 | −0.1645 |
| 6 | 0.0953 | 0.3919 | −0.3118 | 0.3992 | 0.7698 | −0.6674 | −0.5810 | −0.1775 | −0.4318 | −0.7450 | 0.1681 |
| 7 | 0.2029 | 0.7125 | −0.5716 | 0.0432 | −0.3829 | 0.6687 | −0.4168 | 0.2658 | 0.2181 | 0.0705 | −0.1781 |
| 8 | −0.3264 | −0.7266 | 0.3055 | 0.0234 | −0.7161 | 0.0598 | −0.5198 | 0.0999 | 0.6816 | −0.4716 | 0.1862 |
| 9 | −0.6917 | 0.3358 | −0.2423 | −0.3921 | 0.5583 | 0.2305 | 0.3588 | −0.6494 | 0.4169 | −0.7451 | 0.1513 |
| 10 | −0.5985 | −0.8256 | −0.0713 | 0.6598 | −0.2951 | −0.1735 | −0.0084 | 0.6206 | 0.4261 | 0.3258 | −0.2165 |
| 11 | 0.7704 | 0.3803 | 0.4634 | 0.6663 | −0.4879 | 0.6356 | 0.4890 | 0.1110 | 0.2620 | −0.2092 | 0.0304 |
| 12 | −0.3404 | −0.5674 | −0.7932 | −0.5772 | −0.4899 | −0.2049 | −0.5343 | 0.2947 | −0.1619 | −0.1266 | 0.0807 |
| 13 | −0.0338 | −0.6166 | −0.3253 | −0.7251 | 0.0677 | 0.6614 | 0.4311 | −0.7018 | −0.6937 | −0.4153 | 0.1659 |
| 14 | −0.2682 | −0.5091 | 0.0480 | 0.3170 | −0.7225 | 0.8834 | −0.3158 | 0.0101 | −0.4100 | −0.1235 | 0.2524 |
| 15 | 0.2565 | −0.1078 | 0.3414 | −0.4316 | −0.2166 | 0.0072 | −0.7935 | 0.5236 | −0.0566 | −0.4636 | −0.0757 |
| 16 | 0.3958 | 0.4307 | −0.0701 | 0.2972 | −0.6833 | 0.5825 | 0.6117 | −0.5569 | −0.6424 | −0.5044 | 0.2795 |
| 17 | 0.0319 | 0.0469 | −0.3511 | 0.0314 | −0.9649 | 0.7494 | 0.4980 | 0.2465 | −0.7343 | 0.5067 | 0.2693 |
| 18 | −0.5768 | −0.2701 | 0.6151 | −0.5232 | 0.0859 | −0.7951 | −0.0321 | 0.0401 | 0.2094 | −0.0761 | 0.3236 |
| 19 | −0.1144 | −0.7453 | −0.0239 | 0.6069 | 0.8351 | −0.2969 | −0.0053 | 0.0238 | −0.1657 | 0.5311 | 0.1203 |
| 20 | −0.3066 | 0.4596 | −0.4190 | 0.0733 | −0.7205 | −0.9337 | −0.1272 | −0.2780 | −0.3781 | 0.2868 | 0.4707 |
| 21 | −0.7033 | −0.3322 | −0.7593 | 0.0392 | −0.3011 | 0.8993 | 0.5743 | −0.6363 | −0.5388 | −0.5341 | 0.1583 |
| 22 | −0.2757 | 0.5471 | −0.7170 | −0.3693 | −0.8787 | 0.6861 | −0.6731 | −0.6895 | 0.0206 | −0.1790 | 0.3536 |
| 23 | 0.0430 | −0.5891 | −0.5039 | −0.0625 | −0.5739 | −0.1442 | −0.3621 | 0.2378 | −0.2223 | −0.0527 | 0.3039 |
| 24 | 0.0180 | 0.8456 | −0.2889 | −0.6364 | 0.6036 | −0.0969 | −0.3055 | 0.2153 | −0.3660 | 0.1361 | 0.1386 |
| 25 | 0.4003 | 0.1114 | 0.5461 | −0.3675 | 0.6447 | −0.0693 | −0.7118 | 0.6243 | −0.0737 | 0.0206 | 0.3567 |
| 26 | 0.2454 | −0.4345 | 0.3552 | 0.4435 | −0.8293 | 0.3049 | −0.3266 | −0.6133 | −0.5094 | −0.0145 | −0.1643 |
| 27 | −0.3604 | 0.1635 | 0.4420 | 0.7140 | 0.8929 | 0.0178 | −0.2349 | −0.6850 | 0.2051 | 0.4970 | 0.1628 |
| 28 | 0.3333 | 0.0583 | 0.3531 | −0.1965 | −0.3599 | 0.5106 | −0.2346 | −0.7218 | 0.7650 | −0.3121 | 0.3366 |
| 29 | −0.4641 | −0.4724 | −0.5901 | 0.2899 | −0.0581 | −0.5267 | −0.3159 | −0.8717 | −0.0177 | −0.0139 | 0.1253 |
| 30 | 0.1067 | −0.4578 | −0.2740 | 0.2515 | −0.7895 | −0.2120 | 0.1888 | −0.3705 | 0.6872 | −0.1911 | 0.3330 |
| 31 | −0.3392 | −0.1498 | 0.3646 | 0.6204 | −0.1839 | 0.6582 | −0.3199 | 0.1551 | 0.3568 | 0.1923 | 0.1583 |
| 32 | 0.4965 | 0.5742 | −0.3691 | 0.1134 | −0.0984 | −0.8393 | −0.8090 | −0.2186 | −0.6202 | 0.1156 | 0.2307 |
| 33 | 0.6700 | 0.2042 | −0.0350 | 0.3204 | 0.3162 | 0.4469 | −0.8651 | 0.5723 | 0.0352 | 0.3410 | −0.2331 |
| 34 | 0.1773 | −0.5900 | 0.4932 | 0.3290 | −0.4802 | −0.3170 | 0.5759 | 0.5560 | −0.6513 | −0.9507 | 0.3155 |
| 35 | −0.2870 | −0.0699 | −0.0866 | −0.2916 | −0.1423 | 0.3922 | 0.7709 | 0.0266 | −0.4360 | 0.1056 | −0.1400 |
| 36 | 0.3352 | −0.3374 | −0.7176 | −0.3203 | 0.6644 | −0.0147 | 0.3215 | 0.2830 | −0.7252 | 0.4149 | 0.3521 |
| 37 | 0.0020 | −0.2064 | −0.5518 | −0.0655 | 0.2919 | −0.6655 | −0.6246 | −0.6007 | 0.7581 | 0.5035 | −0.1730 |
| 38 | 0.7213 | −0.0464 | 0.2616 | −0.7370 | 0.0446 | −0.4197 | −0.6358 | −0.5214 | −0.4455 | 0.3504 | 0.1515 |
| 39 | 0.0745 | 0.7063 | 0.3609 | 0.0959 | −0.1461 | −0.2016 | −0.1824 | 0.7324 | 0.1838 | −0.7503 | 0.2346 |
| 40 | −0.7435 | −0.4516 | 0.6738 | −0.1484 | −0.1391 | 0.3766 | 0.5637 | 0.3620 | −0.5014 | −0.2568 | 0.1497 |

**Table 12.** Neurons, weights, and biases for the second hidden layer of the ANN model in this study.

*Second Hidden Layer*

| Neuron | First Hidden Layer | | | | | | | | | | | | | | | | | | | |
| | 1 | 2 | 3 | 4 | 5 | 6 | 7 | 8 | 9 | 10 | 11 | 12 | 13 | 14 | 15 | 16 | 17 | 18 | 19 | 20 |
|---|---|---|---|---|---|---|---|---|---|---|---|---|---|---|---|---|---|---|---|---|
| 1 | −0.1406 | −0.5220 | 0.2620 | 0.0433 | 0.0436 | 0.2505 | 0.0772 | 0.3252 | 0.4780 | −0.2089 | 0.1207 | 0.3259 | −0.0076 | −0.3707 | 0.2342 | 0.1582 | −0.4062 | 0.2941 | 0.2141 | −0.4805 |
| 2 | −0.0641 | 0.3851 | −0.0597 | 0.2090 | −0.2911 | 0.2493 | 0.0592 | 0.4414 | 0.1089 | 0.0300 | 0.0559 | −0.1437 | 0.0879 | 0.1158 | 0.1559 | 0.3365 | 0.3371 | 0.2150 | −0.0560 | −0.0775 |
| 3 | 0.1139 | 0.3216 | −0.0013 | 0.2787 | 0.0364 | 0.1226 | −0.4049 | 0.2608 | 0.1083 | −0.2419 | 0.1292 | −0.1896 | −0.0055 | −0.2848 | 0.2669 | 0.2151 | 0.0622 | 0.4145 | −0.0195 | −0.2681 |
| 4 | −0.3921 | 0.2468 | −0.2404 | 0.0857 | −0.1646 | −0.1200 | −0.2069 | −0.2492 | 0.0629 | −0.2256 | −0.3930 | 0.0401 | −0.0819 | −0.3141 | 0.0236 | 0.1730 | −0.2725 | −0.3016 | −0.0239 | −0.3218 |
| 5 | −0.3081 | 0.2905 | −0.2567 | 0.2731 | −0.2699 | −0.0662 | 0.0145 | −0.4032 | −0.3600 | −0.2934 | 0.0785 | 0.2710 | −0.0830 | 0.1712 | −0.0867 | 0.3642 | −0.0871 | −0.4488 | −0.1178 | 0.1331 |
| 6 | 0.0524 | 0.5694 | 0.3378 | −0.0142 | −0.1306 | 0.3685 | 0.1986 | 0.0699 | −0.3546 | 0.3959 | 0.2108 | 0.3006 | 0.3608 | 0.0919 | 0.2979 | −0.1522 | 0.1566 | 0.3083 | 0.2956 | 0.1799 |
| 7 | 0.1532 | 0.4936 | 0.1777 | −0.0361 | −0.3961 | 0.0101 | 0.0091 | 0.3864 | 0.2972 | −0.2380 | 0.1046 | −0.2154 | −0.1241 | 0.3974 | −0.0636 | −0.0613 | 0.3664 | 0.0290 | 0.4316 | −0.1164 |
| 8 | −0.2971 | 0.2172 | −0.4004 | −0.3254 | 0.2207 | −0.0646 | −0.0109 | 0.3193 | 0.3355 | 0.2138 | 0.1669 | −0.0251 | −0.1417 | −0.0079 | −0.1347 | −0.1352 | −0.1244 | −0.1763 | −0.2312 | 0.0903 |
| 9 | −0.2667 | −0.5043 | −0.4362 | 0.2141 | 0.1282 | 0.2850 | 0.4435 | 0.0080 | 0.3238 | −0.4930 | −0.3327 | −0.1548 | −0.1791 | 0.2022 | −0.3028 | −0.0363 | −0.1057 | −0.1486 | 0.1679 | −0.0762 |
| 10 | −0.3028 | −0.0350 | −0.0890 | −0.0354 | 0.3422 | 0.2009 | −0.2660 | 0.0356 | −0.1812 | 0.2848 | −0.1420 | 0.3929 | −0.2222 | 0.1075 | −0.0936 | −0.2289 | 0.3026 | 0.1279 | 0.0528 | −0.0947 |
| 11 | 0.2781 | −0.1349 | 0.0426 | 0.0235 | −0.2321 | −0.1963 | −0.1786 | 0.1776 | −0.4179 | −0.3884 | 0.0947 | −0.4087 | 0.0784 | −0.1527 | 0.1976 | 0.1873 | 0.3724 | 0.4615 | 0.3817 | 0.4288 |
| 12 | −0.3314 | 0.1522 | −0.0697 | −0.1476 | 0.2824 | −0.4825 | −0.3862 | −0.2801 | −0.2798 | −0.4228 | 0.1474 | −0.2675 | −0.2088 | 0.0903 | −0.1468 | 0.0241 | −0.0179 | 0.0285 | 0.2227 | |
| 13 | −0.0243 | −0.3137 | 0.3234 | 0.3165 | 0.2108 | −0.0599 | −0.1065 | −0.2375 | 0.0897 | 0.2386 | 0.2943 | −0.3956 | 0.4071 | −0.1538 | 0.1853 | 0.0392 | 0.1325 | −0.2416 | 0.2460 | −0.3155 |
| 14 | −0.4207 | −0.4424 | −0.4699 | 0.2896 | −0.1649 | 0.1521 | −0.1741 | 0.0379 | −0.2452 | −0.0617 | −0.2094 | −0.3117 | 0.3083 | 0.3793 | 0.3327 | 0.0777 | −0.4512 | −0.0947 | 0.2239 | 0.1765 |
| 15 | 0.1307 | 0.1257 | 0.0679 | −0.2171 | −0.1956 | 0.0543 | 0.1855 | 0.1408 | −0.0027 | −0.2647 | −0.4571 | −0.0934 | 0.0649 | 0.3730 | −0.0351 | −0.2535 | 0.4531 | 0.1238 | −0.0314 | 0.1276 |
| 16 | 0.1312 | 0.1767 | 0.2067 | −0.3377 | −0.1273 | 0.4508 | −0.0290 | −0.2301 | 0.2660 | −0.0051 | 0.2071 | −0.2144 | −0.0116 | −0.0501 | −0.3595 | 0.3049 | 0.0258 | 0.1592 | −0.1188 | 0.4361 |
| 17 | 0.0100 | 0.5449 | 0.3225 | 0.3089 | 0.1688 | 0.0145 | 0.2309 | 0.3835 | 0.2629 | −0.2570 | 0.2010 | −0.1687 | 0.3562 | 0.2249 | −0.3358 | 0.2508 | −0.2291 | 0.0774 | −0.2082 | 0.3325 |
| 18 | −0.2277 | −0.0677 | −0.0486 | −0.3494 | −0.3043 | 0.2349 | −0.1302 | −0.0291 | 0.2950 | −0.4888 | 0.2906 | −0.0258 | 0.2462 | 0.2469 | −0.4652 | 0.1856 | 0.0874 | 0.2405 | −0.3172 | −0.2777 |
| 19 | −0.1043 | 0.0572 | −0.0568 | 0.2316 | 0.0118 | 0.4908 | 0.1115 | 0.2316 | 0.1884 | 0.1855 | −0.1505 | −0.3323 | −0.3209 | −0.1386 | −0.1029 | 0.2485 | −0.1091 | 0.1514 | 0.4495 | 0.0753 |
| 20 | −0.2704 | −0.3722 | 0.2123 | 0.0795 | −0.1884 | 0.1651 | −0.4748 | −0.3302 | −0.1993 | −0.0650 | −0.2008 | 0.2241 | −0.1425 | 0.0771 | 0.0132 | −0.2348 | −0.1603 | −0.1179 | 0.0771 | 0.0409 |
| 21 | −0.4568 | 0.2047 | 0.0152 | −0.3129 | −0.4263 | 0.1829 | −0.0536 | 0.0342 | −0.3056 | 0.0135 | 0.2356 | −0.3022 | −0.1676 | 0.0394 | 0.3372 | 0.4071 | 0.1740 | −0.3581 | −0.0536 | −0.0808 |
| 22 | −0.0709 | −0.1581 | −0.2812 | 0.3837 | −0.1724 | 0.3719 | 0.3878 | 0.0189 | −0.2325 | −0.2006 | 0.4298 | −0.3183 | −0.3252 | 0.1380 | −0.0273 | −0.0010 | 0.1012 | 0.0870 | 0.3157 | −0.1116 |
| 23 | 0.0786 | 0.2550 | −0.3996 | 0.1617 | 0.4286 | −0.3293 | 0.2945 | −0.6878 | −0.0589 | 0.0038 | 0.3619 | −0.5059 | −0.3676 | −0.1545 | −0.0067 | 0.1836 | −0.0754 | −0.2292 | 0.0969 | 0.0506 |
| 24 | 0.0422 | 0.0068 | −0.1938 | 0.1053 | −0.2720 | 0.0101 | −0.0283 | 0.2374 | −0.2937 | −0.0122 | 0.1374 | 0.3433 | −0.1765 | 0.2998 | −0.0573 | 0.2604 | 0.1388 | 0.2053 | −0.2033 | 0.5564 |
| 25 | 0.2520 | 0.3857 | −0.3463 | −0.3316 | −0.2794 | −0.3172 | −0.0038 | −0.0044 | −0.4220 | −0.1891 | −0.2214 | −0.4116 | 0.0366 | −0.0049 | −0.0745 | 0.1595 | 0.0171 | 0.1823 | −0.0339 | 0.3030 |
| 26 | 0.0248 | 0.3386 | 0.0091 | 0.0098 | −0.2635 | 0.4802 | −0.0837 | −0.0436 | 0.0693 | 0.3055 | 0.1646 | 0.3271 | 0.3730 | 0.0243 | −0.4007 | 0.3458 | −0.1571 | 0.2863 | −0.1073 | 0.4638 |
| 27 | −0.1794 | −0.3931 | 0.3003 | −0.4865 | −0.3378 | 0.2500 | 0.3262 | −0.2257 | −0.0038 | −0.1156 | −0.0936 | −0.0671 | −0.2760 | −0.2080 | 0.4135 | 0.3486 | −0.2006 | 0.2483 | 0.2931 | −0.1503 |
| 28 | 0.1610 | 0.2878 | 0.2982 | −0.1795 | 0.2950 | 0.0791 | −0.3599 | −0.1901 | −0.1720 | 0.3546 | 0.3006 | 0.0699 | 0.1269 | 0.1191 | 0.0898 | 0.4007 | 0.0054 | −0.2089 | 0.2814 | 0.3455 |
| 29 | −0.2899 | −0.0251 | 0.1417 | 0.1799 | 0.2286 | −0.0092 | −0.2084 | −0.2463 | −0.2812 | 0.1181 | −0.3383 | −0.0005 | −0.3666 | 0.1527 | 0.2171 | −0.0763 | 0.1146 | −0.2853 | 0.3364 | −0.1187 |
| 30 | 0.0222 | 0.0368 | 0.3488 | 0.3453 | 0.1315 | −0.2404 | 0.2383 | −0.0613 | −0.2496 | −0.0408 | 0.1619 | 0.0431 | −0.2558 | 0.0675 | 0.1443 | 0.4499 | 0.4638 | 0.2888 | 0.0531 | |
| 31 | −0.0876 | −0.3058 | −0.3741 | −0.1372 | 0.2921 | 0.0369 | −0.2710 | 0.0701 | −0.0215 | −0.1588 | 0.3393 | −0.4207 | 0.3685 | 0.1215 | −0.3683 | 0.1407 | −0.0506 | −0.3704 | 0.2876 | 0.0111 |
| 32 | 0.2372 | 0.1311 | −0.1026 | 0.0616 | −0.1637 | 0.1571 | −0.1361 | 0.3294 | −0.1954 | 0.0055 | −0.0174 | −0.2463 | 0.2527 | −0.2722 | −0.4205 | 0.1578 | 0.0866 | −0.0876 | 0.4029 | 0.4013 |
| 33 | 0.3074 | 0.5796 | 0.2672 | 0.1706 | 0.0452 | 0.2762 | −0.1420 | −0.2147 | 0.0023 | 0.1806 | −0.1328 | −0.0393 | 0.3985 | 0.1586 | 0.1846 | 0.2657 | 0.2756 | 0.4651 | −0.2641 | 0.4316 |
| 34 | 0.1582 | 0.4017 | 0.1298 | 0.2009 | −0.2780 | −0.2077 | −0.1437 | −0.2995 | −0.3896 | 0.0455 | 0.0544 | 0.3745 | −0.0589 | 0.2888 | −0.2097 | 0.0206 | 0.1857 | −0.2535 | −0.5150 | 0.1893 |
| 35 | −0.2777 | −0.1416 | −0.0215 | −0.0303 | 0.1246 | −0.1345 | 0.0066 | −0.2876 | 0.4103 | −0.0667 | 0.1351 | −0.2339 | 0.0424 | −0.1074 | −0.1454 | 0.0655 | −0.5144 | 0.1860 | −0.3041 | −0.1637 |
| 36 | 0.0803 | 0.0987 | 0.4278 | 0.1621 | 0.0003 | 0.4169 | 0.1084 | −0.4187 | 0.3001 | 0.2790 | −0.2786 | 0.1928 | −0.2995 | −0.1375 | −0.4427 | −0.2169 | 0.1860 | −0.0640 | −0.0608 | 0.0502 |
| 37 | −0.0223 | 0.1187 | 0.1217 | 0.2629 | 0.2363 | −0.2881 | −0.1345 | −0.0720 | 0.0416 | −0.1344 | 0.1771 | 0.1962 | −0.2916 | 0.2605 | −0.2993 | −0.1979 | 0.1469 | −0.0677 | 0.2635 | −0.0695 |
| 38 | 0.3396 | 0.1540 | −0.2236 | −0.0463 | −0.1797 | 0.2892 | −0.3870 | 0.2219 | 0.2963 | −0.0250 | 0.2840 | 0.3957 | 0.1636 | −0.0511 | −0.3449 | −0.2259 | 0.1768 | −0.0775 | 0.1079 | 0.0958 |
| 39 | 0.0267 | 0.2729 | −0.1904 | −0.2475 | −0.3583 | 0.0162 | −0.3899 | −0.1279 | 0.1726 | −0.1347 | −0.1325 | 0.0407 | −0.1970 | −0.0160 | −0.4419 | −0.3002 | 0.1439 | 0.2575 | 0.0403 | −0.0307 |

**Table 12.** *Cont.*

| | | First Hidden Layer | | | | | | | | | | | | | | | | | | | |
| Neuron | 21 | 22 | 23 | 24 | 25 | 26 | 27 | 28 | 29 | 30 | 31 | 32 | 33 | 34 | 35 | 36 | 37 | 38 | 39 | 40 | Bias |
|---|---|---|---|---|---|---|---|---|---|---|---|---|---|---|---|---|---|---|---|---|---|
| 1 | 0.0768 | −0.2115 | 0.0147 | −0.0994 | −0.0585 | −0.3543 | −0.1715 | 0.2062 | 0.3290 | 0.2921 | −0.5487 | −0.3978 | 0.5125 | −0.1825 | −0.0264 | −0.0328 | 0.3035 | −0.1804 | −0.2091 | 0.1079 | −0.1644 |
| 2 | 0.4092 | 0.5371 | 0.3804 | 0.0179 | 0.0640 | 0.3294 | −0.3631 | −0.2145 | −0.1189 | 0.3614 | −0.0715 | 0.3624 | 0.1609 | 0.4641 | 0.1006 | 0.4432 | −0.2340 | −0.3445 | 0.3538 | −0.0251 | 0.4134 |
| 3 | −0.3651 | −0.2190 | −0.0987 | 0.4580 | 0.2708 | 0.2857 | 0.2903 | 0.3870 | −0.2020 | −0.1909 | 0.1149 | −0.1917 | 0.2763 | −0.2033 | −0.4119 | 0.4892 | −0.1086 | 0.1483 | 0.0987 | 0.0931 | 0.2474 |
| 4 | 0.1350 | −0.2927 | 0.2175 | 0.1051 | −0.2861 | −0.0382 | −0.3253 | 0.3285 | 0.1030 | −0.1720 | 0.3702 | −0.2329 | −0.1797 | 0.1661 | 0.0731 | 0.0200 | −0.0478 | 0.0874 | 0.3169 | 0.1547 | −0.0280 |
| 5 | −0.2425 | 0.0364 | 0.0037 | −0.2898 | −0.1407 | 0.1771 | 0.2238 | −0.2577 | −0.4197 | 0.1141 | 0.2041 | −0.2762 | 0.0760 | 0.1236 | 0.0119 | 0.2809 | −0.1828 | −0.1005 | −0.3647 | 0.3120 | −0.0463 |
| 6 | 0.4142 | 0.1872 | 0.2236 | 0.3866 | −0.1688 | −0.3020 | 0.2444 | 0.3114 | −0.1808 | 0.2120 | 0.1345 | −0.2245 | −0.3553 | −0.0980 | −0.2091 | 0.3503 | 0.1050 | 0.3917 | 0.3177 | 0.0107 | 0.3189 |
| 7 | 0.0223 | 0.4778 | 0.3185 | 0.1326 | −0.2195 | −0.2555 | −0.4893 | −0.2962 | 0.1763 | 0.5453 | −0.0360 | −0.2170 | 0.2171 | 0.1882 | −0.4124 | 0.2948 | −0.2496 | 0.0910 | −0.0572 | 0.2494 | 0.3130 |
| 8 | 0.3333 | 0.0980 | 0.3247 | −0.2698 | −0.3194 | −0.0073 | 0.1240 | −0.4326 | −0.1819 | −0.0623 | −0.3973 | 0.0911 | 0.1929 | −0.0298 | 0.3843 | 0.0455 | −0.3282 | −0.2059 | −0.2972 | −0.3177 | −0.1035 |
| 9 | 0.3598 | 0.3110 | 0.2687 | −0.3894 | −0.0076 | 0.2168 | −0.2589 | −0.0210 | −0.0376 | −0.5393 | −0.2733 | −0.1299 | 0.2692 | 0.0982 | −0.0028 | 0.2771 | −0.1642 | 0.3826 | −0.4205 | 0.0409 | −0.1543 |
| 10 | −0.0953 | 0.0665 | 0.3227 | −0.3548 | −0.2585 | 0.1379 | −0.4123 | −0.2009 | −0.2568 | −0.2654 | 0.0734 | 0.2714 | 0.3240 | 0.1576 | −0.1206 | 0.1251 | 0.3158 | −0.3566 | −0.3726 | 0.2681 | −0.2014 |
| 11 | 0.0458 | 0.0744 | −0.1277 | −0.2280 | 0.0205 | −0.1742 | 0.3784 | −0.0088 | −0.1059 | 0.0576 | 0.1523 | 0.3100 | −0.2735 | 0.3601 | −0.2291 | 0.1558 | −0.4212 | −0.0450 | −0.1154 | 0.3918 | 0.3126 |
| 12 | 0.2147 | 0.1505 | 0.2628 | 0.0269 | −0.1339 | 0.3442 | 0.2972 | −0.1315 | 0.3019 | −0.0895 | −0.1776 | 0.0536 | −0.3029 | −0.1803 | 0.0692 | −0.0582 | 0.3272 | −0.3341 | −0.3433 | −0.2139 | −0.1349 |
| 13 | 0.3632 | −0.2730 | −0.2979 | −0.3590 | −0.2084 | −0.1768 | 0.3498 | 0.4246 | 0.1995 | 0.2020 | 0.0999 | 0.1912 | −0.3977 | 0.2876 | −0.1727 | −0.4491 | −0.2983 | 0.3381 | −0.0804 | 0.2067 | −0.1565 |
| 14 | 0.1828 | 0.3223 | −0.2847 | −0.1042 | −0.0538 | −0.0545 | −0.1139 | −0.0173 | 0.1147 | 0.1866 | −0.2594 | −0.1400 | 0.0958 | −0.2323 | 0.2645 | −0.0352 | 0.0840 | 0.2827 | −0.1351 | −0.2243 | −0.1283 |
| 15 | 0.3233 | 0.3846 | 0.2227 | 0.2052 | 0.1877 | 0.0643 | 0.2911 | 0.2369 | 0.3335 | 0.3638 | −0.0743 | −0.0556 | −0.4286 | 0.4726 | −0.2963 | 0.4134 | 0.3043 | 0.0039 | 0.0831 | −0.2960 | 0.3516 |
| 16 | 0.1076 | 0.4335 | −0.1987 | 0.3943 | 0.0108 | 0.1632 | 0.0260 | −0.2115 | −0.1259 | 0.4290 | 0.1836 | 0.0185 | −0.0354 | 0.0629 | −0.4009 | 0.3964 | 0.3355 | −0.2840 | −0.0638 | −0.3073 | 0.3290 |
| 17 | 0.0945 | −0.0016 | 0.0906 | 0.1655 | 0.3999 | −0.1266 | 0.1329 | −0.2400 | 0.0474 | −0.0254 | −0.3284 | 0.3026 | −0.4537 | 0.1651 | 0.2115 | −0.0801 | −0.2859 | 0.3218 | −0.1546 | 0.4495 | 0.3774 |
| 18 | 0.0656 | 0.0566 | −0.4382 | −0.0410 | 0.0271 | 0.1031 | 0.1107 | 0.4151 | −0.1047 | −0.1403 | −0.2433 | −0.3944 | 0.3803 | −0.0117 | 0.2807 | 0.2279 | 0.3073 | −0.2450 | 0.2814 | 0.2454 | −0.1655 |
| 19 | −0.2544 | −0.3225 | 0.2706 | 0.2289 | 0.1437 | 0.1439 | 0.2801 | 0.3142 | 0.0274 | −0.2264 | −0.3394 | −0.1081 | −0.4067 | 0.3922 | −0.3898 | 0.2621 | −0.2371 | −0.1866 | −0.0031 | 0.2040 | 0.3144 |
| 20 | 0.3372 | 0.1532 | 0.3816 | 0.1402 | 0.0152 | 0.4948 | −0.3293 | −0.4918 | 0.4262 | −0.5084 | −0.5253 | 0.3098 | 0.3345 | −0.1641 | −0.3705 | −0.1531 | 0.2976 | −0.1324 | −0.3610 | 0.0824 | −0.0927 |
| 21 | −0.3999 | −0.4606 | −0.3476 | 0.0986 | 0.0051 | 0.0378 | −0.0709 | −0.0559 | −0.2388 | 0.4424 | 0.1879 | −0.4341 | 0.0372 | −0.0603 | 0.1180 | 0.0371 | −0.1459 | 0.0330 | 0.3200 | 0.2854 | −0.1162 |
| 22 | −0.1028 | −0.3448 | −0.1991 | −0.1618 | 0.2872 | −0.3268 | 0.0497 | −0.3798 | −0.2699 | −0.4961 | 0.2881 | −0.3615 | −0.0233 | −0.2507 | 0.3174 | −0.2758 | 0.2484 | 0.0631 | 0.3159 | −0.2641 | −0.0715 |
| 23 | −0.3099 | −0.0873 | 0.0028 | 0.1242 | −0.6426 | −0.2623 | −0.2061 | 0.0927 | −0.0629 | −0.4159 | −0.0286 | −0.4137 | 0.1579 | −0.3793 | −0.0194 | −0.1521 | 0.3298 | −0.3829 | −0.0159 | −0.1715 | −0.0265 |
| 24 | −0.2692 | 0.3070 | 0.0665 | −0.2796 | −0.1868 | 0.4225 | 0.0645 | 0.3148 | 0.3030 | 0.3923 | 0.0418 | −0.1556 | −0.1325 | −0.1062 | −0.2433 | 0.1793 | −0.3519 | 0.2869 | 0.2944 | −0.2146 | 0.3654 |
| 25 | −0.0114 | −0.3027 | 0.3130 | −0.3195 | 0.2117 | −0.3045 | −0.3834 | −0.2386 | −0.4193 | 0.2101 | 0.0036 | 0.0073 | 0.3075 | −0.0898 | −0.0688 | −0.0589 | −0.0459 | −0.1477 | −0.1263 | −0.0533 | 0.0114 |
| 26 | 0.1722 | 0.2609 | −0.0387 | 0.0130 | 0.1732 | 0.3656 | −0.0309 | 0.1643 | 0.3881 | 0.3819 | 0.2843 | −0.2041 | −0.3372 | −0.1243 | −0.2628 | 0.1515 | −0.0367 | −0.3337 | −0.2197 | 0.1892 | 0.3510 |
| 27 | −0.0566 | −0.3334 | 0.3504 | −0.3445 | 0.0224 | 0.2836 | 0.2706 | −0.0897 | −0.1879 | −0.1814 | −0.4275 | 0.0805 | 0.0043 | 0.1290 | 0.2662 | −0.0484 | 0.2067 | 0.2821 | 0.0961 | 0.3103 | −0.1659 |
| 28 | 0.0355 | 0.2943 | 0.3186 | 0.0362 | 0.0576 | −0.0247 | −0.1003 | 0.0155 | 0.2863 | 0.1684 | 0.1270 | 0.2510 | −0.0391 | 0.0321 | 0.0459 | −0.0124 | 0.0469 | −0.3546 | 0.3117 | 0.2399 | 0.3114 |
| 29 | −0.0427 | 0.0004 | −0.3161 | −0.2563 | 0.0967 | −0.1734 | 0.2595 | −0.1448 | −0.1880 | −0.0135 | −0.1192 | −0.0035 | −0.3545 | 0.1559 | 0.0771 | −0.5105 | 0.2219 | −0.1221 | −0.1376 | −0.3183 | −0.0795 |
| 30 | −0.2031 | 0.1744 | −0.1012 | 0.0341 | 0.2824 | −0.3173 | 0.2654 | 0.2101 | −0.1753 | 0.3096 | 0.0469 | 0.1292 | −0.4504 | 0.2142 | 0.0907 | −0.2426 | −0.2603 | −0.1283 | −0.2757 | 0.4494 | 0.3747 |
| 31 | −0.1771 | −0.3055 | −0.2837 | −0.3544 | 0.1830 | −0.0731 | 0.2997 | 0.4541 | −0.0374 | 0.2708 | 0.1544 | 0.1015 | 0.1715 | 0.1272 | 0.1744 | −0.2146 | 0.1630 | 0.2823 | 0.2248 | −0.4986 | −0.1474 |
| 32 | 0.2211 | −0.4100 | −0.0402 | −0.2369 | 0.3211 | −0.4620 | −0.3468 | −0.4386 | 0.2452 | 0.1687 | −0.2917 | −0.1681 | 0.2985 | 0.3626 | 0.1712 | −0.0989 | 0.1732 | −0.0771 | −0.2158 | 0.0572 | 0.2211 |
| 33 | 0.1280 | −0.0678 | 0.4240 | 0.2189 | 0.3553 | 0.3110 | 0.3399 | 0.3294 | −0.0337 | 0.0175 | −0.0358 | 0.2161 | 0.2083 | −0.1373 | 0.0923 | 0.1798 | −0.0126 | 0.3294 | 0.3085 | 0.0494 | 0.2954 |
| 34 | −0.1181 | 0.4813 | 0.2657 | −0.2980 | −0.1540 | 0.0114 | −0.5148 | 0.3089 | 0.0464 | 0.2665 | 0.1773 | −0.2325 | −0.0546 | −0.2807 | 0.0767 | −0.1897 | −0.0663 | −0.0438 | 0.3282 | 0.1695 | 0.2822 |
| 35 | 0.4581 | 0.3894 | 0.1499 | −0.2715 | −0.2068 | −0.1187 | −0.1987 | −0.2565 | 0.3042 | −0.1692 | 0.0147 | 0.3637 | 0.1131 | 0.2892 | 0.2375 | 0.3863 | 0.3747 | 0.0534 | 0.0752 | 0.2193 | −0.1500 |
| 36 | −0.3805 | −0.2995 | 0.2911 | 0.1375 | 0.3437 | −0.3888 | 0.2640 | 0.0043 | −0.1504 | 0.4632 | −0.3192 | −0.1903 | 0.1491 | 0.4511 | 0.1718 | −0.0885 | 0.2726 | 0.3631 | 0.1532 | −0.2430 | 0.2411 |
| 37 | −0.1288 | 0.4878 | 0.3445 | −0.0369 | 0.1648 | −0.1732 | 0.1582 | −0.2456 | 0.2495 | 0.0618 | 0.1647 | 0.3431 | −0.2958 | 0.1673 | 0.0529 | −0.1069 | −0.2629 | 0.0394 | 0.2848 | 0.4427 | 0.3552 |
| 38 | 0.1515 | 0.2402 | −0.2384 | 0.3824 | 0.0711 | −0.1920 | 0.3547 | 0.2359 | −0.0921 | 0.1478 | 0.0240 | 0.0551 | −0.2322 | 0.1581 | 0.0075 | 0.0301 | −0.2963 | 0.2191 | −0.0286 | 0.1005 | 0.3033 |
| 39 | −0.0981 | −0.3711 | 0.0208 | 0.3721 | 0.2402 | −0.1475 | −0.4202 | −0.2950 | 0.0904 | −0.1111 | 0.0284 | 0.3176 | −0.0301 | 0.4134 | −0.0569 | 0.4671 | −0.1343 | −0.0426 | 0.3297 | 0.0781 | 0.2472 |

*(Second Hidden Layer — row label for Table 12)*

**Table 13.** Neurons, weights, and biases for the output layer for the ANN model in this study.

| | Second Hidden Layer Neuron | | | | | | | | | | | | | | | | | | | |
| | 1 | 2 | 3 | 4 | 5 | 6 | 7 | 8 | 9 | 10 | 11 | 12 | 13 | 14 | 15 | 16 | 17 | 18 | 19 | 20 |
|---|---|---|---|---|---|---|---|---|---|---|---|---|---|---|---|---|---|---|---|---|
| Weight | −0.2425 | 0.1079 | 0.1664 | −0.3607 | −0.4238 | 0.1753 | 0.3360 | −0.2893 | −0.3836 | −0.2694 | 0.4022 | −0.1757 | −0.0361 | −0.2227 | 0.3160 | 0.2980 | 0.1107 | −0.2892 | 0.1082 | −0.4448 |

| | Second Hidden Layer Neuron | | | | | | | | | | | | | | | | | | | |
| | 21 | 22 | 23 | 24 | 25 | 26 | 27 | 28 | 29 | 30 | 31 | 32 | 33 | 34 | 35 | 36 | 37 | 38 | 39 | Bias |
|---|---|---|---|---|---|---|---|---|---|---|---|---|---|---|---|---|---|---|---|---|
| Weight | −0.3494 | −0.2602 | −0.1228 | 0.1338 | 0.2542 | 0.1623 | −0.1214 | 0.2947 | −0.2579 | 0.1706 | −0.1471 | 0.4089 | 0.4170 | 0.2894 | −0.2602 | 0.1802 | 0.0809 | 0.4036 | 0.3339 | 0.2847 |

*8.2. ANN Model with Optimized Input Parameters*

Previously, the ANN model with the provided inputs resulted in high-accuracy prediction; however, the model required many of experimental data. In an effort to make the ANN model more robust and faster, a further study was performed on reducing the number of inputs to decrease the number of experimental input data as well as speed up the training processes. In the model, the temperatures were unchanged because they directly have impact on the p$K_a$ values. This study focuses on the omission of some inputs to obtain an acceptable trade-off between the errors, complexities, and training speed.

From the previous full model, the study reduces the number of inputs such as density, refractive index, and the number of atoms (molecular weights) or combinations and examines the $R_{overall}$, $MSE_{train}$, $MSE_{validation}$, and $MSE_{test}$. The results are summarized in Table 14 below. The $R_{overall}$ reached the highest values of 0.89225, while $MSE_{train}$, $MSE_{validation}$, and $MSE_{test}$ were 0.00881, 0.02945, and 0.0798, respectively, when the sound velocity and refractive index were omitted. By comparison with the full model, the value of $R_{overall}$ of the full model was a little higher than the $R_{overall}$ of the reduced model, while the values of $MSE_{train}$, $MSE_{validation}$, and $MSE_{test}$ of the full model were lower than the values of the reduced model. However, the differences were not large, and the reduced model did not require two sets of experimental datasets, which were the speed of sound and refractive index. Other combinations and the results are shown in Table 14.

**Table 14.** Comparisons of the reduced ANN models' performance.

| Removed Inputs | $R_{overall}$ | $MSE_{train}$ | $MSE_{validation}$ | $MSE_{test}$ |
|---|---|---|---|---|
| Density | 0.8165 | 0.0139 | 0.1127 | 0.0882 |
| Refractive Index | 0.8295 | 0.0020 | 0.6125 | 0.1453 |
| Sound Velocity | 0.8425 | 0.0127 | 0.1369 | 0.0369 |
| Viscosity | 0.6939 | 0.0578 | 0.1931 | 0.1629 |
| Molecular Weight (MW) | 0.7976 | 0.0317 | 0.0973 | 0.1132 |
| Sound Velocity & MW | 0.8416 | 0.0053 | 0.0740 | 0.0903 |
| Refractive Index & MW | 0.8321 | 0.0084 | 0.0847 | 0.0905 |
| Density & MW | 0.7231 | 0.0058 | 0.1476 | 0.1427 |
| Density & Sound Velocity | 0.8871 | 0.0105 | 0.0661 | 0.0532 |
| Density & Refractive Index | 0.6270 | 0.0367 | 0.2419 | 0.1563 |
| Density & Viscosity | 0.8141 | 0.0038 | 0.1202 | 0.0782 |
| Refractive Index & Sound Velocity | 0.8922 | 0.0088 | 0.0295 | 0.0789 |
| Refractive Index & Viscosity | 0.6970 | 0.0028 | 0.1146 | 0.1953 |
| Sound Velocity & Viscosity | 0.7965 | 0.0080 | 0.1086 | 0.1048 |

Tables 15–17 summarize the weights and biases for the first, second hidden, and output layers of the reduced (refractive index and sound velocity) model.

**Table 15.** Neurons, weights, and biases for the first hidden layer for the reduced (refractive index and sound velocity) ANN model.

| Neuron | Temperature | Molecular Weight | Number of C | Number of H | Number of N | Number of O | Density | Viscosity | Bias |
|---|---|---|---|---|---|---|---|---|---|
| 1 | −0.1124 | 0.6348 | −0.1873 | 0.4645 | 0.1034 | 0.2243 | 0.0002 | −0.1052 | −0.1158 |
| 2 | 0.0615 | −0.3450 | −0.8294 | −0.1796 | −0.4868 | −0.2349 | 0.0533 | −0.3133 | 0.1328 |
| 3 | 0.5461 | −0.0927 | 0.6757 | 0.2881 | −0.6074 | 0.4297 | −0.4679 | −0.5559 | 0.1199 |
| 4 | 0.0927 | 0.5841 | −0.1074 | 0.5233 | −0.4246 | −0.4336 | −0.4754 | 0.5572 | −0.1090 |
| 5 | 0.1490 | −0.5444 | 0.6037 | 0.4292 | 0.4622 | −0.4950 | −0.2201 | −1.0798 | 0.3012 |
| 6 | 0.5757 | 0.2660 | −0.5993 | 0.0854 | −0.4168 | 0.5109 | −0.3404 | 0.8664 | 0.2324 |
| 7 | 0.5141 | 0.5379 | 0.4001 | 0.5429 | −0.2441 | 0.4389 | −0.5126 | 0.3049 | 0.1068 |
| 8 | 0.0563 | 0.2174 | 0.4328 | 0.5608 | 0.5297 | −0.1892 | −0.3629 | 0.4120 | 0.2365 |
| 9 | 0.4652 | −0.3000 | 0.1158 | 0.7321 | 0.3856 | −0.8809 | 0.3438 | 0.4824 | −0.0531 |
| 10 | 0.7418 | 0.3283 | −0.6438 | 0.7119 | 0.4064 | −0.8005 | −0.1244 | −0.3977 | 0.1852 |
| 11 | 0.2276 | 0.8759 | 0.6164 | 0.8747 | 0.0400 | 0.3592 | −0.8716 | 0.6410 | −0.2100 |
| 12 | 0.1087 | −0.7280 | 0.2788 | 0.0503 | −1.2021 | −0.0328 | 0.3785 | −0.0224 | 0.3546 |
| 13 | −0.5202 | 0.6770 | −0.0261 | −0.6915 | 0.8760 | −0.9841 | 0.0394 | 0.6508 | 0.1744 |
| 14 | 0.0531 | 0.8788 | 0.9902 | −0.0819 | −0.3536 | −0.1227 | −0.2712 | 0.7150 | 0.1844 |
| 15 | 0.2421 | 0.8376 | −0.6455 | 0.9184 | 0.6953 | 0.4072 | −0.2738 | 0.8454 | −0.1895 |
| 16 | 0.0890 | 0.4068 | 0.0635 | 0.5081 | 0.5596 | −0.4588 | −0.4789 | −0.8583 | 0.2146 |
| 17 | 0.2639 | −0.3405 | −0.6563 | −0.5088 | 0.1728 | 0.3697 | −0.4624 | −0.4943 | 0.1529 |
| 18 | 0.7855 | −0.3594 | 0.5913 | 0.1127 | 0.3087 | 0.1566 | 0.6119 | −0.3837 | 0.1080 |
| 19 | 0.1410 | −0.0114 | 0.7488 | −0.7060 | −0.6965 | 0.3823 | 0.6775 | −0.2402 | 0.2199 |
| 20 | 0.3099 | −0.6274 | −0.3260 | −0.1632 | −0.4577 | −0.4790 | −0.7153 | 0.0169 | 0.3601 |
| 21 | 0.4944 | 0.0026 | 0.5356 | −0.0617 | −0.0308 | 0.5477 | −0.6159 | −0.7886 | 0.3109 |
| 22 | 0.7639 | −0.1264 | 0.2089 | 0.9107 | −0.5838 | −0.7296 | 0.6672 | −0.2214 | −0.2152 |
| 23 | 0.1166 | 0.1045 | 0.3700 | −0.9411 | 0.8753 | 0.2054 | 0.3032 | 0.1617 | −0.0329 |
| 24 | 0.0609 | −0.1350 | −0.1921 | 0.3794 | −0.8310 | 0.6281 | 0.6988 | −0.6927 | 0.2667 |
| 25 | −0.6552 | 0.7355 | −0.4250 | 0.3771 | 0.2129 | −0.3710 | −0.0342 | −0.6665 | 0.3866 |
| 26 | 0.6890 | −0.5657 | 0.6196 | −0.2864 | −0.3520 | −0.2700 | 0.5165 | 0.5083 | −0.1347 |
| 27 | −0.6732 | 0.2030 | 0.5096 | −0.1973 | −0.9718 | −0.6391 | 0.6424 | −0.6936 | 0.2777 |
| 28 | 0.2173 | 0.5484 | −0.3094 | 0.0621 | 0.1741 | −0.6550 | 0.2073 | 0.4688 | −0.0303 |
| 29 | −0.6974 | 0.4656 | 0.1492 | −0.3026 | 0.7717 | 0.9050 | −0.7429 | −0.6089 | 0.1536 |
| 30 | 0.1526 | −0.4806 | −0.6957 | 0.8162 | −0.7643 | 0.4780 | 0.2927 | 0.7566 | 0.1915 |
| 31 | 0.0945 | −0.5134 | −0.2309 | 0.7205 | 0.9501 | −0.5224 | −0.5596 | 0.6869 | 0.0859 |
| 32 | 0.2465 | −0.2837 | −0.2653 | 0.6383 | −0.7153 | −0.4952 | −0.4650 | −0.5650 | 0.5163 |
| 33 | 0.0365 | −0.7152 | 0.6098 | 0.8256 | −0.3674 | −0.8255 | −0.1359 | 0.8580 | −0.2322 |
| 34 | 0.5208 | 0.2592 | −0.8057 | 0.1775 | 0.1458 | 0.3018 | −0.5660 | 0.3805 | −0.3196 |
| 35 | −0.2938 | −0.7255 | 0.7986 | 0.3319 | −0.0357 | 0.0326 | 0.2618 | 0.4753 | 0.2338 |

**Table 15.** *Cont.*

| Neuron | Temperature | Molecular Weight | Number of C | Number of H | Number of N | Number of O | Density | Viscosity | Bias |
|---|---|---|---|---|---|---|---|---|---|
| 36 | 0.0461 | −0.5900 | 0.9098 | −0.7344 | −0.6309 | 0.5015 | −0.2017 | 0.5709 | 0.3059 |
| 37 | 0.1191 | 0.5190 | 0.5997 | −0.7229 | −0.3856 | 0.8398 | −0.3775 | −0.2760 | 0.2416 |
| 38 | 0.9021 | 0.4589 | −0.7767 | 0.2780 | −0.1635 | 0.1610 | −0.3135 | 0.6061 | −0.2489 |
| 39 | −0.2666 | −0.0804 | 0.1347 | 0.1864 | −0.7957 | 0.4831 | −0.7491 | −0.6082 | 0.6132 |
| 40 | 0.2850 | 0.7118 | −0.7793 | −0.2098 | −0.5247 | 0.3116 | −0.1219 | 0.4998 | −0.1063 |

**Table 16.** Neurons, weights, and biases for the second hidden layer of the reduced (refractive index and sound velocity) ANN model.

| Neuron | 1 | 2 | 3 | 4 | 5 | 6 | 7 | 8 | 9 | 10 | 11 | 12 | 13 | 14 | 15 | 16 | 17 | 18 | 19 | 20 |
|---|---|---|---|---|---|---|---|---|---|---|---|---|---|---|---|---|---|---|---|---|
| 1 | −0.1130 | 0.2500 | 0.3210 | −0.2440 | −0.0862 | 0.2500 | 0.1120 | 0.1610 | −0.0157 | −0.0831 | 0.0158 | −0.0696 | −0.1380 | −0.5570 | −0.2110 | −0.3910 | 0.0779 | −0.0925 | 0.2760 | 0.2720 |
| 2 | 0.1920 | 0.0169 | −0.0515 | −0.0243 | 0.2800 | 0.1490 | 0.0796 | 0.0199 | −0.0004 | −0.3660 | 0.0374 | −0.2410 | 0.0160 | −0.0773 | 0.1210 | 0.1660 | −0.4440 | −0.3050 | 0.2660 | −0.0588 |
| 3 | 0.3380 | 0.0986 | −0.2280 | 0.1810 | 0.0651 | 0.2510 | −0.1890 | −0.2970 | −0.0560 | −0.3270 | 0.1470 | −0.3060 | −0.0109 | 0.0110 | −0.0519 | 0.3530 | 0.0070 | −0.1130 | 0.0289 | 0.0852 |
| 4 | 0.0029 | −0.2870 | −0.1520 | 0.2410 | −0.0852 | −0.2720 | −0.3080 | 0.2290 | 0.0400 | −0.3450 | −0.1940 | −0.2340 | −0.2220 | 0.2550 | −0.2780 | −0.1950 | −0.2800 | −0.3630 | 0.1030 | 0.2020 |
| 5 | −0.0007 | 0.1900 | −0.3000 | 0.3990 | 0.0803 | −0.0425 | −0.0978 | 0.0110 | −0.1940 | −0.1910 | 0.2480 | −0.1630 | −0.4170 | 0.3800 | −0.0309 | 0.1470 | 0.3210 | −0.0858 | −0.3130 | 0.0712 |
| 6 | 0.0756 | −0.3330 | 0.3130 | −0.3870 | 0.3590 | 0.0350 | 0.3970 | 0.3560 | −0.0578 | 0.1760 | 0.3720 | 0.3200 | 0.2500 | 0.1930 | −0.2960 | −0.2860 | 0.2310 | 0.0960 | −0.1210 | 0.0331 |
| 7 | 0.1040 | −0.2420 | 0.4010 | 0.2160 | 0.0436 | −0.0560 | 0.1370 | 0.2910 | −0.0003 | 0.2710 | −0.3060 | −0.0497 | 0.1310 | −0.1280 | 0.3250 | −0.3770 | 0.2800 | 0.1590 |  | 0.2920 |
| 8 | −0.4420 | −0.0948 | 0.2920 | 0.1810 | 0.4170 | 0.0836 | −0.3010 | 0.2410 | 0.4190 | −0.2300 | 0.3710 | −0.3380 | −0.1050 | 0.3020 | −0.1650 | 0.1830 | −0.0523 | 0.2230 | 0.1380 | −0.4550 |
| 9 | −0.2180 | 0.0290 | 0.3720 | 0.0618 | −0.2620 | 0.1110 | 0.4140 | −0.2540 | −0.2210 | −0.0907 | −0.0221 | 0.0773 | −0.4180 | −0.0377 | 0.0335 | 0.0189 | −0.5090 | 0.2020 | 0.3530 | −0.3040 |
| 10 | −0.2720 | 0.2590 | −0.1100 | −0.1360 | 0.2760 | 0.2640 | −0.1120 | −0.2390 | −0.2080 | 0.1240 | 0.4520 | 0.3210 | 0.2330 | −0.0212 | 0.0116 | −0.1730 | −0.3350 | 0.1470 | −0.1530 | 0.3590 |
| 11 | −0.2510 | 0.3300 | −0.2370 | −0.0599 | 0.1830 | −0.2230 | −0.0796 | −0.2410 | −0.0792 | 0.1490 | 0.0378 | 0.3650 | −0.2550 | −0.4200 | −0.0699 | 0.3110 | −0.3300 | −0.4470 | −0.3490 | 0.1100 |
| 12 | 0.0316 | 0.3330 | −0.3090 | −0.1350 | 0.0516 | 0.2210 | 0.0025 | −0.2220 | 0.3730 | −0.1260 | −0.1940 | −0.2900 | −0.1520 | 0.3920 | 0.0601 | −0.1390 | 0.2470 | 0.0685 | −0.0674 | 0.1090 |
| 13 | −0.0388 | −0.3170 | −0.2070 | −0.1660 | 0.2590 | −0.2280 | −0.2220 | −0.1330 | 0.1220 | 0.3330 | −0.1190 | 0.1020 | 0.0698 | 0.1670 | 0.1320 | 0.1160 | −0.0554 | −0.2800 | 0.1730 | −0.0591 |
| 14 | 0.0190 | 0.0555 | 0.1490 | 0.1250 | 0.0242 | −0.3140 | 0.0072 | 0.1240 | −0.0191 | 0.2440 | 0.0171 | −0.1740 | −0.3590 | −0.4420 | 0.2790 | −0.4100 | −0.0249 | −0.3060 | 0.0589 | −0.2580 |
| 15 | −0.2480 | −0.1440 | 0.1710 | 0.0332 | 0.2420 | −0.3640 | 0.2500 | 0.3490 | 0.2740 | −0.2760 | −0.1150 | −0.1000 | 0.1400 | −0.3710 | −0.2810 | −0.0389 | −0.0917 | −0.4830 | −0.0861 | −0.3190 |
| 16 | 0.2560 | −0.3850 | −0.1730 | −0.2310 | −0.2540 | −0.2530 | 0.4930 | 0.0831 | −0.1310 | −0.3690 | −0.1120 | −0.3630 | −0.3610 | −0.1680 | −0.0180 | 0.1160 | 0.2210 | 0.0690 | −0.1470 | −0.4690 |
| 17 | −0.0384 | −0.3590 | 0.5390 | −0.2420 | −0.3240 | −0.0394 | 0.2010 | 0.3080 | −0.4700 | 0.2240 | 0.0091 | 0.0401 | 0.0514 | 0.1080 | −0.2060 | 0.0955 | −0.4210 | 0.3750 | −0.0885 | 0.2980 |
| 18 | −0.2950 | −0.3000 | −0.1890 | 0.1410 | −0.2440 | 0.0490 | −0.2770 | −0.2220 | −0.2540 | −0.1050 | 0.2530 | −0.3600 | 0.0608 | 0.1560 | −0.3110 | 0.1850 | −0.1350 | 0.3540 | 0.0164 | 0.4220 |
| 19 | −0.2430 | −0.3460 | 0.0064 | −0.0890 | 0.1500 | −0.1740 | 0.3440 | −0.2930 | 0.3920 | −0.1010 | 0.2510 | 0.0933 | −0.1920 | −0.2670 | −0.2180 | −0.0743 | −0.0221 | −0.2540 | 0.0888 | −0.0590 |
| 20 | 0.3380 | −0.1120 | 0.0249 | −0.2560 | −0.2260 | 0.0755 | −0.0957 | 0.0723 | 0.1660 | −0.3140 | −0.1740 | −0.1030 | 0.2620 | −0.0429 | 0.3520 | −0.0546 | −0.5240 | −0.3140 | −0.2880 | 0.2930 |
| 21 | 0.3230 | −0.2700 | −0.1200 | 0.3670 | −0.1290 | 0.2630 | 0.1230 | 0.0128 | 0.1540 | 0.1810 | 0.3660 | −0.2640 | 0.0532 | 0.2870 | 0.2850 | −0.3410 | −0.0020 | −0.1740 | −0.3190 | −0.0187 |
| 22 | 0.0352 | −0.0088 | 0.5040 | −0.2230 | −0.0909 | −0.0749 | 0.3040 | −0.1270 | −0.1930 | −0.2020 | −0.1280 | 0.1080 | 0.3320 | −0.0513 | 0.0652 | −0.4480 | −0.1390 | 0.2900 | −0.3190 | 0.4650 |
| 23 | 0.1490 | −0.0417 | −0.0190 | 0.0193 | 0.1130 | 0.4670 | −0.3390 | −0.0389 | −0.6830 | 0.1560 | −0.2640 | 0.3310 | 0.1060 | −0.5160 | −0.0698 | −0.1700 | 0.0358 | 0.0629 | −0.3860 | 0.3450 |
| 24 | −0.2800 | 0.3840 | −0.2310 | 0.2820 | −0.0917 | −0.3810 | 0.1120 | −0.4170 | 0.4020 | 0.2990 | −0.1880 | 0.1770 | 0.1130 | −0.1740 | 0.3190 | 0.4320 | −0.1500 | −0.1340 | 0.1130 | −0.3120 |
| 25 | 0.1340 | −0.2330 | 0.3780 | 0.1480 | 0.3840 | 0.0049 | 0.4240 | 0.3490 | −0.1530 | 0.0334 | 0.1990 | 0.0184 | 0.1930 | 0.4150 | 0.0104 | −0.1710 | −0.2410 | −0.2550 | 0.2260 | −0.0698 |
| 26 | 0.1170 | −0.0838 | 0.2580 | −0.1580 | −0.1960 | 0.2170 | 0.0862 | 0.1630 | 0.0054 | −0.1030 | −0.3430 | 0.1080 | 0.3290 | 0.0529 | −0.2430 | −0.0290 | 0.1090 | −0.2080 | 0.0738 | −0.0798 |
| 27 | −0.0954 | −0.2500 | −0.2370 | 0.0241 | 0.2720 | −0.4600 | −0.2940 | −0.0971 | 0.0194 | 0.2640 | 0.3480 | −0.3600 | 0.2010 | 0.1520 | −0.0396 | 0.1520 | 0.2660 | −0.2790 | −0.0808 | −0.1420 |
| 28 | −0.4080 | −0.0986 | 0.1700 | −0.0138 | 0.1070 | −0.3740 | −0.0988 | 0.1680 | −0.3900 | −0.2110 | 0.2860 | −0.4180 | 0.2430 | 0.4610 | 0.3830 | −0.2890 | −0.1340 | −0.1910 | −0.1510 | −0.3600 |
| 29 | 0.2320 | −0.1360 | 0.0196 | −0.1240 | −0.2440 | −0.1990 | 0.3110 | 0.0540 | 0.2630 | 0.1390 | 0.0372 | −0.0138 | 0.1140 | −0.3000 | −0.3360 | 0.3680 | −0.0042 | 0.2600 | −0.3650 | 0.2610 |
| 30 | −0.2010 | −0.3090 | 0.0304 | 0.1500 | −0.1350 | −0.0872 | 0.0809 | −0.0779 | 0.3430 | −0.1970 | 0.2830 | −0.0369 | 0.2690 | −0.1400 | −0.3710 | 0.3700 | 0.3680 | 0.0276 | 0.0356 | 0.0485 |
| 31 | −0.0109 | 0.2620 | 0.0684 | −0.3730 | 0.3830 | −0.3660 | 0.1570 | 0.0693 | −0.3550 | −0.1110 | 0.0335 | 0.0252 | 0.0903 | −0.2630 | −0.1930 | 0.3330 | 0.2560 | −0.2560 | −0.0864 | 0.0487 |
| 32 | 0.1300 | 0.1450 | 0.2550 | −0.1580 | 0.3840 | 0.2800 | 0.0257 | 0.3190 | 0.3320 | −0.3570 | 0.0169 | 0.0013 | −0.2150 | 0.2470 | −0.3410 | 0.3160 | −0.3590 | −0.3670 | −0.0673 | −0.2200 |
| 33 | 0.1370 | −0.3300 | 0.5440 | −0.3770 | −0.1580 | 0.0080 | 0.2370 | 0.1800 | −0.5010 | −0.1450 | −0.4210 | 0.0773 | 0.0540 | −0.3490 | −0.1650 | −0.3110 | −0.1160 | 0.2390 | −0.0323 | −0.1220 |
| 34 | 0.3330 | 0.1900 | 0.1750 | −0.1040 | −0.3110 | 0.3150 | −0.2790 | −0.4850 | −0.0676 | 0.1020 | 0.3610 | −0.2340 | 0.1930 | −0.2300 | −0.2990 | 0.2350 | 0.0459 | 0.0252 | 0.3210 | −0.1690 |
| 35 | −0.2650 | −0.3780 | 0.2790 | 0.0695 | −0.0657 | −0.1310 | 0.0574 | 0.1130 | 0.2480 | 0.0132 | 0.0252 | −0.4200 | −0.0322 | 0.1710 | 0.2310 | −0.2350 | 0.3410 | −0.3970 | −0.0309 | −0.5530 |
| 36 | −0.4190 | 0.1270 | 0.2730 | −0.2380 | 0.2420 | 0.1230 | 0.4570 | 0.0449 | −0.1700 | 0.0361 | 0.4880 | −0.2730 | −0.2990 | −0.1060 | −0.1830 | 0.0858 | −0.0591 | 0.1280 | −0.2980 | 0.0320 |
| 37 | 0.1960 | −0.3640 | −0.3330 | 0.3330 | −0.0504 | 0.0437 | −0.1210 | −0.0168 | −0.2780 | 0.4350 | −0.2780 | 0.0938 | −0.3890 | −0.2170 | 0.2900 | 0.3750 | 0.3900 | 0.1480 | 0.0185 | 0.2540 |
| 38 | −0.1250 | 0.1690 | 0.3140 | −0.3080 | 0.3440 | −0.1110 | −0.3820 | −0.1420 | −0.0428 | 0.1510 | 0.3200 | −0.0004 | −0.3230 | −0.2000 | 0.3560 | −0.2130 | −0.2230 | −0.1290 | −0.1020 | −0.2820 |
| 39 | 0.2530 | −0.3750 | 0.1610 | −0.3010 | 0.4360 | 0.0415 | 0.1960 | 0.1800 | 0.1160 | −0.0007 | 0.2630 | −0.0127 | −0.8470 | 0.3670 | −0.2930 | 0.1230 | 0.0762 | 0.1090 | −0.2980 | 0.2700 |

**Table 16.** *Cont.*

| | | | | | | | | | | First Hidden Layer | | | | | | | | | | | |
|---|---|---|---|---|---|---|---|---|---|---|---|---|---|---|---|---|---|---|---|---|---|
| Neuron | 21 | 22 | 23 | 24 | 25 | 26 | 27 | 28 | 29 | 30 | 31 | 32 | 33 | 34 | 35 | 36 | 37 | 38 | 39 | 40 | Bias |
| 1 | −0.2730 | 0.0241 | 0.0586 | 0.1230 | −0.3640 | 0.4180 | −0.0802 | 0.3160 | −0.2580 | 0.2790 | 0.0292 | 0.0445 | 0.0115 | 0.2910 | 0.0121 | 0.4050 | −0.1890 | 0.2620 | 0.2830 | −0.2870 | −0.1300 |
| 2 | −0.1890 | −0.2880 | −0.1730 | −0.2920 | −0.4680 | 0.0963 | 0.1120 | −0.1130 | −0.2070 | −0.0012 | −0.3620 | 0.1670 | 0.1280 | 0.3690 | −0.0649 | −0.1080 | 0.0911 | −0.3640 | 0.1360 | 0.0472 | −0.0060 |
| 3 | 0.0824 | −0.3320 | 0.0834 | 0.0689 | 0.2140 | 0.2460 | −0.3180 | −0.1520 | 0.0215 | 0.2540 | −0.1190 | −0.2150 | 0.1340 | 0.2970 | −0.2680 | 0.0341 | −0.1290 | −0.1840 | −0.3580 | 0.2840 | −0.2810 |
| 4 | 0.0893 | 0.2420 | 0.0307 | 0.3120 | 0.1910 | 0.3050 | 0.3240 | 0.3570 | 0.1880 | 0.3640 | −0.2310 | −0.2110 | −0.2420 | −0.0009 | −0.2650 | 0.2470 | −0.2180 | −0.3790 | 0.2430 | −0.2340 | 0.1110 |
| 5 | 0.2660 | −0.3340 | 0.1300 | −0.3480 | 0.3930 | −0.3470 | −0.3070 | 0.4240 | 0.2150 | 0.3310 | 0.2440 | 0.2470 | −0.0926 | −0.2160 | 0.1730 | 0.0244 | 0.5080 | −0.0909 | 0.2340 | −0.0719 | 0.3590 |
| 6 | 0.2710 | 0.0494 | −0.3220 | −0.3120 | 0.0197 | −0.0076 | −0.1170 | −0.0111 | −0.2800 | 0.1310 | 0.3040 | 0.3420 | −0.0285 | 0.2520 | 0.2470 | −0.3650 | 0.2370 | −0.5280 | 0.2230 | 0.3720 | −0.2220 |
| 7 | −0.0358 | −0.0607 | 0.2880 | 0.1010 | −0.1510 | 0.1890 | −0.0261 | −0.0277 | 0.3520 | −0.2860 | 0.3980 | 0.2860 | −0.0266 | 0.2570 | −0.4420 | −0.3840 | −0.0780 | −0.0569 | 0.2790 | 0.1100 | 0.3390 |
| 8 | 0.1120 | −0.0708 | 0.1320 | 0.2400 | −0.2110 | −0.0409 | 0.2030 | 0.2640 | −0.2500 | 0.1240 | 0.2240 | 0.1500 | −0.2300 | −0.1210 | −0.2290 | 0.1440 | −0.2230 | −0.0033 | 0.3100 | 0.3630 | −0.1380 |
| 9 | −0.2270 | 0.2440 | 0.1670 | 0.0033 | −0.1700 | −0.3710 | −0.1400 | 0.1080 | −0.2710 | 0.2220 | 0.1310 | 0.1720 | −0.3310 | 0.4130 | 0.1710 | 0.0056 | 0.2040 | −0.0890 | 0.0505 | 0.1320 | 0.4180 |
| 10 | 0.0282 | −0.2740 | −0.0571 | −0.0259 | −0.3390 | 0.2080 | 0.3370 | 0.4490 | 0.2060 | 0.2130 | −0.0665 | −0.1800 | −0.2940 | −0.2830 | −0.0382 | 0.1600 | 0.3900 | −0.4600 | 0.2140 | 0.3370 | −0.0764 |
| 11 | −0.0297 | 0.2340 | −0.1780 | 0.2200 | 0.2170 | 0.0996 | −0.1510 | 0.1020 | −0.2890 | −0.1040 | 0.4440 | −0.3110 | −0.2700 | 0.1840 | −0.1570 | 0.1150 | −0.2330 | −0.1200 | −0.2480 | 0.1060 | 0.3660 |
| 12 | 0.3220 | −0.2130 | 0.4240 | −0.3430 | 0.0036 | 0.2010 | −0.3630 | 0.2500 | 0.0003 | 0.5270 | 0.0604 | −0.1120 | 0.0187 | 0.0065 | 0.0328 | −0.3520 | 0.5280 | −0.0426 | 0.1910 | 0.2680 | −0.0493 |
| 13 | 0.3570 | 0.3450 | 0.3840 | −0.1080 | 0.5010 | 0.3540 | −0.2540 | 0.3480 | 0.4640 | 0.1770 | 0.2810 | −0.0030 | 0.0358 | 0.2670 | −0.0289 | −0.2560 | −0.1920 | 0.1600 | 0.3160 | 0.3140 | −0.2050 |
| 14 | −0.0115 | 0.4120 | −0.0639 | −0.0970 | 0.1810 | 0.3550 | −0.3820 | −0.3680 | 0.0968 | −0.2860 | −0.1490 | 0.2160 | −0.1550 | 0.2590 | −0.5380 | −0.1340 | 0.1890 | −0.1370 | −0.1760 | −0.3930 | 0.4180 |
| 15 | 0.0546 | −0.2080 | −0.2710 | 0.2860 | −0.2560 | −0.1530 | −0.2860 | −0.1730 | −0.3360 | 0.1330 | 0.1880 | −0.0471 | 0.2890 | −0.1130 | −0.1450 | 0.1470 | −0.2460 | 0.1130 | −0.2760 | −0.2520 | −0.0389 |
| 16 | −0.1790 | −0.4350 | −0.2260 | 0.0623 | −0.1380 | −0.2760 | 0.1010 | 0.3770 | 0.4480 | 0.3390 | −0.2290 | 0.4560 | −0.1200 | 0.4730 | 0.3230 | 0.2110 | −0.1630 | 0.0827 | 0.1550 | −0.1650 | 0.4360 |
| 17 | −0.0218 | 0.2120 | −0.0201 | 0.2200 | −0.0996 | −0.0491 | 0.1340 | −0.0855 | 0.0624 | −0.0366 | −0.2980 | −0.0259 | −0.0883 | 0.3030 | 0.0039 | −0.2320 | 0.1990 | 0.1650 | 0.2060 | 0.1050 | −0.0284 |
| 18 | −0.1460 | −0.0002 | 0.2940 | 0.1470 | −0.3310 | 0.0808 | −0.0047 | 0.2240 | 0.3140 | 0.1410 | −0.0268 | −0.3220 | 0.2880 | 0.2940 | 0.0685 | −0.0824 | −0.2120 | 0.0935 | 0.2750 | −0.0016 | −0.0134 |
| 19 | −0.1430 | 0.3190 | 0.4000 | −0.4270 | −0.0802 | −0.3670 | −0.1450 | 0.3060 | 0.0910 | −0.3340 | 0.2900 | −0.1600 | −0.2540 | 0.1960 | 0.3460 | −0.0326 | 0.0948 | 0.0627 | 0.1100 | −0.0681 | −0.0467 |
| 20 | −0.0757 | 0.2630 | 0.4100 | 0.1670 | −0.0939 | 0.3280 | 0.0689 | −0.1040 | 0.2590 | 0.2390 | −0.1910 | −0.0578 | 0.0517 | 0.3660 | 0.2950 | −0.5640 | 0.2400 | −0.2340 | 0.2460 | 0.0994 | −0.2430 |
| 21 | −0.3380 | 0.3370 | 0.2970 | 0.2730 | 0.3100 | 0.1590 | −0.1330 | 0.2090 | 0.2090 | −0.2960 | −0.3620 | 0.2770 | −0.0921 | 0.2440 | 0.2370 | −0.4540 | 0.4140 | −0.0264 | 0.3810 | 0.1650 | 0.3660 |
| 22 | −0.0518 | 0.2050 | −0.0306 | 0.3290 | 0.0575 | −0.3510 | 0.0622 | 0.1970 | −0.1130 | −0.1160 | −0.1040 | −0.2270 | 0.3280 | −0.2470 | 0.0266 | 0.1070 | −0.4230 | 0.0395 | −0.3440 | −0.0499 | −0.2410 |
| 23 | 0.0831 | 0.1980 | −0.0057 | 0.1180 | 0.4100 | 0.1190 | 0.2250 | −0.0615 | −0.4070 | 0.2060 | −0.0358 | −0.3850 | −0.2600 | 0.1350 | −0.3720 | 0.0546 | 0.0049 | 0.3020 | −0.3770 | −0.3620 | 0.3520 |
| 24 | 0.1650 | −0.3540 | −0.0870 | 0.1900 | 0.1660 | −0.3150 | −0.3580 | 0.2960 | 0.1360 | 0.2370 | 0.1390 | 0.3780 | 0.2800 | −0.0392 | 0.2460 | −0.1910 | −0.1820 | −0.3840 | 0.1600 | 0.0449 | −0.1320 |
| 25 | 0.0142 | −0.1400 | 0.0656 | 0.1770 | 0.1780 | 0.0130 | −0.2910 | −0.0520 | 0.2430 | 0.1050 | −0.0088 | 0.1500 | −0.2160 | 0.4340 | 0.0893 | 0.0348 | 0.0881 | 0.0479 | 0.1570 | −0.0406 | −0.0833 |
| 26 | 0.1350 | 0.1630 | 0.0975 | 0.1040 | −0.2940 | 0.2350 | −0.0977 | −0.0799 | 0.2770 | −0.3640 | 0.1580 | 0.0133 | −0.2110 | 0.0938 | 0.0750 | −0.1290 | −0.2430 | 0.0005 | 0.0342 | 0.1580 | 0.2940 |
| 27 | 0.2840 | 0.1460 | −0.2810 | −0.0268 | −0.1660 | 0.2460 | −0.0308 | 0.2130 | 0.2660 | 0.0197 | 0.2720 | 0.4390 | −0.1730 | 0.4530 | 0.0600 | −0.1360 | 0.2340 | −0.3820 | −0.0679 | 0.4550 | −0.0389 |
| 28 | −0.0367 | 0.3100 | −0.0807 | 0.1320 | −0.3480 | 0.3960 | −0.0883 | −0.2700 | −0.2210 | −0.0311 | −0.0597 | 0.0086 | 0.1270 | 0.2940 | 0.2920 | 0.3870 | 0.4010 | −0.1280 | −0.2870 | 0.3280 | 0.4150 |
| 29 | 0.1880 | 0.2810 | −0.1410 | −0.3250 | −0.3330 | 0.3290 | −0.2420 | −0.3820 | −0.1540 | 0.1030 | −0.5220 | 0.0084 | −0.0021 | −0.0271 | −0.3290 | −0.0818 | 0.2060 | 0.1660 | 0.2080 | −0.1210 | 0.3650 |
| 30 | −0.2180 | −0.3470 | 0.4060 | −0.0490 | −0.1130 | −0.0974 | 0.2960 | −0.2990 | −0.1380 | 0.0404 | 0.1530 | 0.3010 | −0.2890 | 0.2020 | 0.0073 | 0.2320 | 0.5000 | 0.0542 | 0.0797 | 0.3700 | 0.3550 |
| 31 | 0.3260 | 0.2310 | 0.0774 | −0.0236 | −0.2220 | −0.2830 | −0.3340 | 0.0838 | 0.1500 | −0.2440 | 0.1750 | 0.1870 | −0.3610 | −0.1330 | 0.3800 | −0.0907 | −0.0522 | 0.2330 | 0.4460 | −0.1470 | −0.0919 |
| 32 | 0.3100 | 0.1300 | 0.0115 | 0.0298 | 0.2330 | −0.2130 | −0.3090 | −0.1380 | 0.2270 | 0.2880 | −0.3850 | 0.2100 | −0.3050 | −0.2590 | 0.1610 | 0.2330 | 0.4720 | 0.0809 | −0.2720 | 0.0171 | 0.4060 |
| 33 | 0.2420 | 0.2010 | 0.2560 | 0.5010 | 0.5700 | −0.4740 | 0.0251 | −0.3590 | −0.4110 | −0.0292 | −0.0418 | −0.3860 | 0.1180 | −0.1380 | −0.2850 | −0.1780 | −0.4230 | −0.2220 | −0.2670 | 0.1840 | −0.0154 |
| 34 | −0.1170 | 0.3720 | −0.1490 | 0.3640 | −0.0036 | 0.0038 | −0.2390 | 0.1490 | −0.2370 | 0.2480 | −0.3150 | −0.1940 | −0.3910 | −0.0069 | −0.0771 | −0.4080 | −0.2070 | −0.1090 | 0.2070 | −0.1420 | 0.3260 |
| 35 | −0.3790 | −0.0276 | 0.1140 | −0.3410 | 0.1120 | 0.3910 | −0.0957 | 0.3670 | 0.3670 | 0.2200 | 0.1070 | −0.2050 | 0.2610 | −0.0786 | 0.1930 | −0.3680 | 0.0401 | −0.4460 | 0.3450 | 0.2160 | 0.4100 |
| 36 | −0.0678 | 0.2650 | 0.0287 | −0.2950 | −0.4210 | 0.4100 | 0.0842 | 0.2690 | 0.2560 | 0.2190 | 0.0226 | −0.0609 | −0.1530 | 0.0761 | 0.4220 | −0.1060 | −0.1370 | −0.1380 | 0.3630 | 0.4110 | −0.0845 |
| 37 | −0.0603 | −0.1800 | 0.1390 | −0.0983 | 0.1170 | 0.0986 | 0.1820 | −0.1960 | −0.0012 | −0.1290 | −0.2580 | 0.4070 | −0.1000 | 0.3660 | −0.1360 | −0.6470 | 0.3620 | −0.1590 | −0.1610 | 0.0724 | 0.4610 |
| 38 | 0.2140 | −0.0106 | −0.3590 | 0.3200 | 0.2470 | 0.0174 | −0.0384 | −0.1490 | −0.3600 | 0.3110 | −0.3310 | −0.1590 | 0.3370 | −0.1420 | −0.0017 | 0.0373 | −0.3480 | 0.3090 | −0.0060 | −0.3920 | −0.1270 |
| 39 | 0.4130 | 0.3740 | 0.0496 | −0.0222 | 0.2280 | −0.0449 | −0.0397 | 0.5130 | 0.2090 | 0.2650 | −0.5730 | 0.0793 | 0.1410 | −0.1630 | −0.2370 | −0.4570 | 0.4560 | 0.0582 | 0.5070 | 0.2690 | 0.3590 |

*(Row label: Second Hidden Layer)*

**Table 17.** Neurons, weights, and biases for the output layer for the reduced (refractive index and sound velocity) ANN model.

| | Second Hidden Layer Neuron | | | | | | | | | | | | | | | | | | | |
|---|---|---|---|---|---|---|---|---|---|---|---|---|---|---|---|---|---|---|---|---|
| | 1 | 2 | 3 | 4 | 5 | 6 | 7 | 8 | 9 | 10 | 11 | 12 | 13 | 14 | 15 | 16 | 17 | 18 | 19 | 20 |
| Weight | −0.3520 | −0.0192 | −0.2240 | 0.2270 | 0.4230 | −0.2280 | 0.4020 | −0.3160 | 0.1970 | −0.4330 | 0.1430 | −0.1250 | −0.3330 | 0.4030 | −0.2460 | 0.1050 | −0.1970 | −0.0096 | −0.3040 | −0.3800 |

| | Second Hidden Layer Neuron | | | | | | | | | | | | | | | | | | | |
|---|---|---|---|---|---|---|---|---|---|---|---|---|---|---|---|---|---|---|---|---|
| | 21 | 22 | 23 | 24 | 25 | 26 | 27 | 28 | 29 | 30 | 31 | 32 | 33 | 34 | 35 | 36 | 37 | 38 | 39 | Bias |
| Weight | 0.3080 | −0.2810 | 0.3210 | −0.0485 | −0.3340 | 0.3540 | −0.3420 | 0.1720 | 0.4340 | 0.3530 | −0.4710 | 0.2820 | −0.3130 | 0.2380 | 0.3790 | −0.2280 | 0.1970 | 0.0030 | 0.2470 | 0.3220 |

Figure S8 shows the performance of the reduced ANN model with several epochs ranging from 140 to 200. The figure shows that the best output of the model was when the number of epochs reached 187.

Figure S9 shows the parity plot between the estimated and the experimental (target) values. The overall R was over 0.89, while Figure S10 represents the error histogram with very few outliers. Based on Figure S10, most of the errors were close to the value of 0.0136 while most of the errors were in the range of −0.0785 to 0.1057.

### 8.3. The Simulated ANN Model with Simulated Inputs

As observed in the previous sections, the ANN inputs included many other experimental measurements, such as densities, viscosities, and so on. It would consume more time to obtain all the measured data than directly measuring the $pK_a$. In addition, it is impossible to obtain all of the data to estimate the dissociation constants values at various temperatures if the chemicals are synthesized. Therefore, it would be more convenient to develop an ANN model that can estimate the dissociation constant values based on the simulated data. Therefore, the model is called the simulated ANN model in this study. In this section, the chemicals are used to obtain the ANN model, the same as in the previous ANN models. In addition to the temperatures (K), the molecular weight (MW), and the output values ($pK_a$) taken from the literature, the simulated data which include critical pressure ($P_c$/kPa), critical temperature ($T_c$/K), acentric factor ($\omega$), the boiling temperature at 1 atm ($T_b$/K), flash point temperature ($T_F$/K), density ($\rho$/g·cm$^{-1}$), dynamic viscosity ($\eta$/cP), and vapor pressure ($P_v$/mbar), were used for the ANN model. In particular, critical pressures, critical temperatures, acentric factors, and boiling temperatures were obtained by the NIST estimation system from Aspen Plus V11.0, while flash point temperatures, densities, dynamic viscosities, and vapor pressures were calculated by CosmothermX (version C30_1201). In summary, the input of the ANN model includes 10 different types of data while the outputs are the same as the previous records. Because of the same number of inputs and outputs, the numbers of neurons for the two hidden layers are kept the same. The ANN structure for the model is the same as in Figure S4. After training, the obtained parameters were $R_{overal}$ = 0.9988, $MSE_{overal}$ = 0.0012, $MSE_{train}$ = 0.0016, $MSE_{validation}$ = $4.8433 \times 10^{-5}$, and $MSE_{test}$ = $1.1246 \times 10^{-4}$. The weights and biases for the first, second hidden, and output layers are reported in Tables 18–20. The simulated ANN model performance with various epochs is shown in Figure S11. Figure S12 shows the parity plot between the estimated and the experimental values, while Figure S13 represents the error histogram of the simulated ANN model.

**Table 18.** Neurons, weights, and biases for the first hidden layer for the simulated ANN model.

| Neuron | Temperature | Molecular Weight | $P_c$ | $T_c$ | $\omega$ | $T_b$ | Flash Point | Density | Viscosity | Vapor Pressure | Bias |
|---|---|---|---|---|---|---|---|---|---|---|---|
| 1 | 0.3791 | −0.8085 | −0.0754 | −0.9132 | 0.3418 | 0.5538 | −0.7291 | −0.6512 | −0.9860 | 0.4810 | −2.0023 |
| 2 | 0.4746 | 0.8302 | 0.8654 | −0.9043 | −0.7836 | −0.3686 | 0.2534 | 0.4265 | −0.6060 | −0.6293 | −1.8939 |
| 3 | −0.9678 | 1.0067 | −0.7232 | 0.7869 | 0.8227 | −0.3581 | −0.4781 | −0.0987 | 0.2129 | −0.0762 | 1.7755 |
| 4 | −0.7234 | −0.5354 | −0.8370 | 0.0832 | −0.9282 | −0.6581 | 0.8251 | −0.2978 | −0.0111 | −0.7344 | 1.7409 |
| 5 | 0.9134 | 0.4450 | 0.7416 | −0.5013 | −0.8255 | 0.3571 | −0.2302 | 0.1854 | 0.8122 | 0.8372 | −1.6066 |
| 6 | −0.2562 | 0.3493 | −0.7441 | 1.1380 | −0.5083 | −0.0830 | 0.2571 | −0.6824 | −0.2094 | 1.2723 | 1.2660 |
| 7 | 0.1311 | 0.9507 | −0.8731 | 0.2632 | −0.7579 | −0.5542 | 0.3877 | 0.0767 | −0.3177 | 1.0348 | −1.3825 |
| 8 | −0.1516 | 0.9403 | −0.8621 | −0.8131 | −0.1601 | 0.6607 | −0.2984 | 0.9383 | −0.4385 | 0.4668 | 1.3223 |
| 9 | −0.8780 | −0.3078 | −0.7529 | 0.6730 | −0.6046 | 0.4587 | 0.1584 | −0.8601 | −0.0597 | 0.9301 | 1.1700 |
| 10 | 0.5599 | 0.2238 | 0.5272 | 1.0303 | −0.0462 | 0.3535 | 0.5984 | 1.0233 | 0.3259 | 0.8472 | −1.0670 |
| 11 | 0.3167 | 0.4763 | 0.7830 | 0.0952 | 0.8615 | −0.1111 | 0.3839 | −0.8183 | −0.7026 | 1.0308 | −1.0445 |
| 12 | −0.1015 | −0.9264 | 0.4596 | 0.7620 | 0.6767 | −0.1300 | 0.7635 | 0.2730 | −0.9578 | −0.5940 | 0.8757 |
| 13 | 0.2397 | 1.1468 | −0.3341 | 0.5968 | 0.1786 | −0.1154 | 1.2483 | −0.7094 | −0.5689 | 0.5591 | −0.6607 |
| 14 | 0.4327 | −0.9618 | 0.6721 | −0.9927 | 0.3691 | −0.0770 | −0.9889 | 0.0254 | −0.4886 | −0.4149 | −0.6969 |
| 15 | 0.1752 | 0.1638 | 0.5640 | 0.2419 | −0.7975 | 1.0040 | 0.6430 | 0.4276 | −0.8962 | −0.4620 | −0.6266 |
| 16 | −0.0787 | −0.5875 | −0.0880 | 0.1842 | 0.3355 | 1.1467 | 1.0754 | 0.4767 | 0.1912 | −0.9335 | 0.5461 |
| 17 | 0.8310 | 0.3786 | −0.7886 | 0.1869 | −0.7642 | 0.2922 | 1.0674 | 0.4767 | 0.0537 | −0.9244 | −0.3521 |
| 18 | −0.1769 | 0.7507 | 0.1771 | 0.9779 | 0.1770 | 0.9286 | −0.9233 | 0.8800 | 0.1105 | 0.0204 | 0.1429 |
| 19 | 0.6280 | −0.1534 | 0.6922 | 0.0460 | 0.7548 | −0.8116 | −1.1484 | 0.7920 | 0.3856 | 0.3359 | −0.2327 |
| 20 | 0.8099 | −0.4914 | 0.2947 | −1.1076 | 0.1261 | −0.8730 | −0.0074 | −0.9000 | 0.4258 | 0.1104 | −0.0017 |
| 21 | 0.6538 | 0.1012 | 0.4083 | 0.9067 | −0.1955 | −0.8266 | −0.7945 | 0.8350 | 0.6301 | 0.7756 | −0.1463 |
| 22 | −0.0648 | −0.4560 | 0.0470 | −0.0373 | 0.9708 | 0.4811 | −0.8173 | −0.4484 | 0.7598 | 1.0971 | −0.1087 |
| 23 | −0.8342 | 0.7138 | −0.2714 | 0.8072 | 0.3474 | −0.6189 | −0.1420 | 0.7397 | −0.5940 | −0.8895 | −0.2606 |
| 24 | −0.4484 | −1.0355 | −0.6716 | −0.7935 | −0.3745 | −0.9331 | −0.5373 | −0.3708 | −0.2612 | −0.3261 | −0.2835 |
| 25 | 0.1710 | −0.9951 | −0.8705 | 0.7688 | 0.4988 | −0.2195 | −0.2865 | −0.8504 | 0.3641 | 0.5624 | 0.5392 |
| 26 | 0.3659 | −0.2618 | −0.2483 | 0.5968 | 0.5172 | 0.3471 | −1.0394 | 0.3535 | 0.7768 | −1.0728 | 0.6771 |
| 27 | 1.0669 | 0.4071 | 0.7614 | 0.2302 | 0.0432 | −1.1352 | 0.3036 | 0.0534 | −1.0864 | −0.4443 | 0.7538 |
| 28 | 0.1242 | −0.5401 | 0.7899 | −0.4472 | 0.6723 | 1.1611 | 0.4083 | −0.8136 | −0.5353 | 0.7056 | 0.7908 |
| 29 | −0.6794 | 0.0376 | −0.4322 | −0.5894 | 0.7935 | −0.6744 | −0.3704 | 0.4906 | −0.7393 | −0.7163 | −1.0006 |
| 30 | −0.4455 | 0.6158 | 0.7486 | 0.9645 | 0.6431 | 0.6781 | −0.5696 | 0.2461 | 0.8275 | 0.4248 | −0.9722 |
| 31 | −0.1922 | 0.2504 | 0.7396 | 0.5626 | 0.8359 | 0.6437 | −0.6817 | 0.9742 | −0.0605 | −0.2694 | −1.1254 |
| 32 | 0.9841 | −0.6248 | 1.3213 | −0.1506 | −0.5872 | −0.2341 | −0.7170 | −0.2421 | −0.1931 | 0.6931 | 1.1572 |
| 33 | −0.3461 | −0.6864 | −0.6441 | −0.1240 | 0.9806 | 0.8088 | −0.4116 | −0.0670 | 0.0843 | 1.1457 | −1.3049 |
| 34 | 0.7468 | 0.5503 | −0.8377 | 0.1024 | 1.0355 | −0.6581 | −0.6267 | −0.6024 | −0.0731 | −0.0491 | 1.4496 |
| 35 | 0.8815 | −0.3559 | −1.0640 | 0.2585 | 0.4074 | −0.0368 | −0.0374 | −0.4715 | 0.9178 | 0.7317 | 1.5280 |
| 36 | −1.0656 | −0.5278 | 0.3542 | 0.5742 | −0.6614 | −0.5160 | 0.7826 | 1.0751 | −0.0855 | −0.2976 | −1.5972 |
| 37 | −0.1119 | −0.1186 | −0.5653 | −0.6578 | −1.1791 | −0.8569 | 0.3319 | −0.7798 | 0.0706 | 0.9059 | −1.6048 |
| 38 | 0.7690 | 0.0281 | −0.2250 | −0.2455 | −0.7056 | 0.4202 | −1.2924 | 0.8117 | −0.1272 | −0.5649 | 1.8023 |
| 39 | 1.1043 | −0.0561 | −0.1044 | 0.8983 | 0.5968 | −0.2814 | 1.1681 | −0.0199 | −0.2836 | 0.2912 | 1.9324 |
| 40 | 0.7479 | 0.1511 | −0.3996 | 0.8378 | −0.3424 | 0.0579 | 0.9685 | −0.7035 | 0.7730 | 0.2840 | 2.1333 |

**Table 19.** Neurons, weights, and biases for the second hidden layer of the simulated ANN model.

| Neuron | 1 | 2 | 3 | 4 | 5 | 6 | 7 | 8 | 9 | 10 | 11 | 12 | 13 | 14 | 15 | 16 | 17 | 18 | 19 | 20 |
|---|---|---|---|---|---|---|---|---|---|---|---|---|---|---|---|---|---|---|---|---|
| 1 | 0.3675 | −0.4129 | −0.3632 | −0.3588 | −0.2176 | −0.0267 | 0.3484 | 0.3196 | −0.1697 | −0.1709 | −0.3583 | 0.3200 | 0.1792 | −0.1927 | −0.2804 | 0.1303 | −0.2889 | −0.1000 | 0.0965 | −0.0488 |
| 2 | 0.0723 | 0.0893 | 0.2059 | 0.2444 | −0.4186 | −0.4020 | −0.3264 | 0.1877 | 0.2626 | 0.2094 | −0.2386 | 0.2902 | −0.1840 | 0.2470 | −0.1751 | −0.1253 | 0.2915 | −0.0251 | 0.0453 | 0.2768 |
| 3 | 0.2520 | −0.0459 | 0.2321 | −0.4832 | 0.3868 | 0.2867 | −0.0423 | 0.0069 | −0.0826 | −0.1255 | −0.1334 | −0.1317 | 0.0078 | 0.1420 | −0.1252 | −0.3232 | 0.1581 | 0.0203 | 0.3040 | −0.0614 |
| 4 | 0.4175 | −0.3582 | 0.0947 | −0.2655 | 0.3440 | 0.1157 | 0.1400 | 0.2081 | −0.2398 | 0.1097 | −0.2520 | −0.1662 | 0.4494 | −0.1493 | 0.1973 | 0.2679 | −0.0214 | −0.0811 | −0.2637 | 0.4300 |
| 5 | −0.0879 | 0.3366 | −0.0763 | −0.3183 | −0.0477 | 0.0366 | −0.0852 | −0.3360 | 0.1690 | 0.0904 | −0.0626 | −0.0554 | −0.3462 | 0.2023 | 0.0354 | 0.3661 | −0.2668 | −0.2026 | −0.3864 | 0.2922 |
| 6 | 0.0209 | 0.3113 | −0.1321 | −0.2340 | −0.2750 | 0.3039 | −0.3866 | 0.3813 | 0.0887 | −0.1052 | −0.2553 | −0.2168 | 0.0243 | −0.0599 | −0.2273 | 0.3987 | −0.2026 | 0.1238 | −0.2777 | 0.0689 |
| 7 | −0.3080 | 0.2053 | −0.3001 | 0.0640 | −0.0090 | 0.3091 | −0.2579 | 0.3353 | −0.1685 | 0.3183 | 0.0014 | 0.1246 | 0.3819 | 0.1811 | 0.2232 | −0.0066 | 0.3930 | 0.2130 | 0.1698 | 0.3030 |
| 8 | −0.0073 | −0.2561 | 0.3986 | −0.1968 | −0.3530 | −0.2169 | −0.2618 | −0.0461 | 0.0599 | −0.2809 | 0.4086 | −0.2643 | −0.3737 | −0.2666 | −0.0633 | 0.1066 | 0.2228 | −0.0179 | −0.2473 | 0.2051 |
| 9 | 0.1161 | −0.0462 | 0.1493 | 0.1134 | −0.2454 | 0.2519 | 0.1486 | 0.2451 | 0.3223 | 0.1135 | −0.2602 | −0.4068 | 0.2248 | −0.1390 | −0.0694 | 0.2290 | 0.3209 | −0.2798 | 0.2943 | −0.0195 |
| 10 | −0.1006 | 0.4132 | −0.0918 | −0.0236 | 0.3483 | 0.1348 | −0.2190 | 0.0887 | −0.1846 | 0.0750 | 0.3235 | −0.2270 | 0.0389 | 0.1166 | −0.2783 | −0.0850 | 0.0414 | −0.0632 | −0.2019 | 0.5208 |
| 11 | 0.2539 | 0.1689 | 0.3385 | 0.0161 | 0.1556 | −0.0100 | 0.0744 | 0.2135 | 0.2024 | 0.0003 | 0.3023 | 0.3141 | 0.2711 | 0.2036 | 0.0149 | 0.2833 | 0.1908 | −0.2238 | 0.3016 | −0.2702 |
| 12 | −0.3592 | −0.0462 | 0.1392 | −0.3342 | −0.2028 | 0.3489 | 0.1980 | −0.2600 | −0.1874 | 0.0458 | −0.3512 | −0.0885 | 0.0658 | −0.3567 | 0.0946 | −0.2922 | 0.2332 | 0.0328 | −0.3001 | −0.1538 |
| 13 | −0.2886 | −0.3729 | −0.1169 | 0.2808 | 0.2341 | 0.1289 | 0.0902 | −0.2957 | 0.3308 | −0.2536 | −0.0484 | 0.2546 | 0.0342 | 0.2909 | 0.2895 | 0.4214 | 0.0593 | 0.3112 | 0.1030 | −0.1277 |
| 14 | 0.0138 | 0.2795 | −0.3605 | −0.0229 | 0.2648 | 0.1849 | −0.2092 | −0.1621 | 0.1922 | 0.0532 | 0.1551 | −0.0228 | 0.2506 | −0.2150 | 0.2870 | 0.2236 | 0.1281 | 0.3172 | −0.1741 | 0.1862 |
| 15 | −0.0233 | −0.1131 | −0.0705 | −0.3079 | 0.1192 | −0.1939 | 0.1198 | −0.2788 | −0.4236 | 0.2974 | 0.2018 | −0.2785 | −0.1920 | 0.3985 | −0.2390 | −0.2483 | 0.1623 | −0.0480 | −0.1519 | 0.3837 |
| 16 | −0.1981 | −0.1014 | 0.0239 | 0.1274 | 0.3778 | 0.3309 | −0.3028 | 0.1922 | −0.3277 | −0.0402 | −0.2290 | −0.0610 | −0.4548 | 0.0082 | −0.0364 | 0.1288 | 0.1384 | 0.2139 | −0.3628 | −0.1981 |
| 17 | −0.0689 | −0.0225 | 0.0828 | 0.0220 | −0.1723 | −0.0796 | −0.3003 | −0.4118 | 0.3329 | −0.2405 | 0.2664 | −0.4247 | 0.3493 | 0.1485 | 0.0030 | 0.1454 | 0.0842 | −0.3032 | 0.4409 | −0.3327 |
| 18 | 0.4089 | −0.1471 | 0.0197 | −0.1433 | 0.3188 | −0.2334 | −0.1666 | 0.3047 | 0.2030 | 0.1801 | 0.3162 | −0.2553 | 0.0271 | 0.1325 | −0.0231 | 0.1256 | −0.2650 | −0.4236 | −0.2432 | −0.2862 |
| 19 | −0.2853 | 0.4080 | −0.0046 | −0.4050 | 0.2562 | −0.1497 | 0.2916 | 0.3208 | −0.1797 | 0.0102 | −0.1906 | −0.0893 | −0.4142 | −0.1010 | 0.0319 | −0.3426 | 0.0227 | −0.0607 | −0.3114 | −0.2433 |
| 20 | −0.3529 | −0.3001 | −0.3084 | −0.0962 | −0.2981 | 0.0110 | −0.2989 | −0.1328 | −0.4082 | −0.0031 | 0.1524 | 0.1811 | 0.1314 | −0.4248 | 0.3186 | −0.0809 | −0.2305 | 0.2504 | −0.2408 | 0.1477 |
| 21 | 0.2171 | −0.0027 | −0.2431 | −0.4023 | −0.3551 | −0.2860 | 0.2289 | −0.2751 | −0.0777 | −0.1006 | −0.1067 | −0.0416 | 0.1949 | 0.1177 | −0.2360 | 0.3695 | −0.0246 | 0.1837 | −0.0999 | 0.1438 |
| 22 | 0.0998 | 0.2137 | 0.3127 | 0.2138 | 0.3085 | −0.3175 | 0.1966 | 0.3467 | −0.1056 | 0.0322 | 0.3431 | −0.1981 | 0.3399 | 0.2871 | −0.1244 | 0.2051 | −0.4745 | −0.1835 | −0.2246 | −0.2419 |
| 23 | 0.1705 | 0.0087 | 0.0597 | 0.1643 | −0.1616 | 0.4684 | 0.3213 | −0.3196 | 0.2598 | 0.2103 | −0.3800 | 0.1234 | 0.4029 | 0.0309 | 0.2855 | −0.2844 | 0.0771 | −0.3007 | −0.2744 | 0.0795 |
| 24 | 0.1822 | 0.0475 | 0.2643 | 0.3493 | 0.1944 | 0.3380 | −0.4132 | 0.2575 | −0.2732 | 0.0306 | −0.2744 | −0.0610 | 0.1496 | 0.0011 | −0.2392 | −0.2243 | 0.3115 | −0.1077 | 0.3326 | 0.3206 |
| 25 | −0.0879 | −0.3220 | −0.0218 | −0.4372 | 0.1675 | 0.4062 | 0.1276 | −0.2029 | 0.3063 | −0.1298 | −0.1556 | −0.3889 | −0.0704 | 0.1230 | −0.0527 | −0.2653 | −0.2306 | 0.1100 | 0.2157 | 0.1574 |
| 26 | 0.0664 | −0.2748 | 0.1905 | −0.4100 | −0.1411 | 0.4253 | −0.1676 | 0.3268 | 0.1000 | 0.2088 | −0.1635 | 0.0874 | 0.3895 | 0.2700 | −0.1899 | −0.2011 | −0.3417 | 0.4476 | 0.3642 | −0.2204 |
| 27 | −0.4107 | −0.0125 | 0.0655 | 0.2521 | −0.3257 | 0.2403 | −0.0277 | 0.1031 | −0.3835 | 0.0320 | 0.3745 | −0.2255 | −0.2633 | 0.1177 | −0.2763 | 0.2297 | 0.3443 | 0.1277 | −0.0795 | −0.1062 |
| 28 | 0.3569 | 0.0092 | 0.3149 | 0.2706 | 0.0290 | 0.1662 | −0.1881 | −0.3940 | −0.3172 | −0.4059 | 0.1257 | −0.2374 | 0.1872 | 0.0076 | −0.5490 | 0.3238 | 0.1654 | −0.3454 | 0.1546 | −0.3176 |
| 29 | −0.1188 | −0.1834 | −0.1737 | −0.1057 | −0.4265 | −0.2526 | 0.3211 | −0.3484 | −0.0458 | −0.2569 | −0.3911 | 0.1499 | −0.4155 | 0.1673 | −0.2108 | −0.0418 | −0.0798 | −0.1626 | 0.0003 | 0.4023 |
| 30 | −0.0506 | −0.0415 | −0.0595 | 0.2957 | −0.1181 | −0.4751 | 0.3638 | −0.2912 | 0.2586 | −0.3027 | 0.0055 | −0.0266 | 0.1468 | 0.1001 | −0.0611 | −0.0706 | −0.1491 | −0.3745 | 0.0767 | −0.4548 |
| 31 | 0.0659 | 0.0940 | 0.1306 | −0.4552 | −0.4220 | −0.0516 | −0.2078 | −0.2641 | 0.2216 | 0.1184 | −0.0141 | −0.5029 | 0.2138 | 0.1098 | −0.1704 | −0.0172 | −0.2573 | 0.1849 | 0.0480 | 0.4035 |
| 32 | −0.1916 | 0.1135 | −0.2770 | 0.2752 | −0.2372 | 0.0198 | −0.4276 | −0.1118 | 0.0289 | −0.4154 | −0.4634 | 0.2876 | −0.3036 | 0.3862 | −0.0791 | −0.2791 | 0.0479 | −0.2018 | 0.1299 | 0.3007 |
| 33 | 0.2503 | 0.3602 | −0.4307 | −0.3453 | 0.3625 | 0.4604 | 0.3820 | 0.1913 | 0.2390 | 0.0332 | −0.2940 | −0.0425 | 0.3309 | 0.0389 | 0.3755 | 0.0181 | 0.2091 | −0.2622 | −0.2552 | −0.2866 |
| 34 | 0.1769 | 0.2671 | −0.0335 | −0.1262 | −0.0452 | 0.3025 | −0.1254 | −0.5197 | 0.2315 | −0.0828 | −0.0241 | 0.2819 | −0.0379 | −0.1626 | 0.3203 | 0.2423 | −0.1577 | 0.2614 | −0.3634 | −0.1480 |
| 35 | 0.1022 | 0.0818 | −0.2743 | 0.0858 | 0.3471 | 0.0006 | 0.1746 | −0.4219 | −0.2315 | 0.2351 | −0.2417 | −0.1334 | −0.2856 | 0.0773 | −0.3517 | −0.1749 | −0.3622 | −0.0222 | −0.2148 | −0.2513 |
| 36 | −0.0260 | 0.3176 | 0.2087 | −0.3842 | −0.1337 | 0.3318 | 0.2631 | 0.2989 | −0.2377 | −0.2938 | 0.0150 | −0.3735 | −0.2342 | 0.1842 | 0.2201 | 0.1427 | −0.2026 | −0.1728 | −0.0199 | −0.1311 |
| 37 | −0.1950 | 0.2480 | 0.2322 | −0.2545 | −0.1468 | 0.1649 | −0.0072 | 0.1848 | 0.4621 | −0.0661 | −0.3462 | −0.2857 | −0.2961 | 0.0959 | −0.0986 | −0.4596 | 0.2270 | −0.1956 | 0.2720 | −0.0417 |
| 38 | 0.2485 | 0.3513 | 0.3696 | 0.0124 | −0.2639 | −0.4020 | −0.0510 | 0.3320 | −0.1695 | −0.2361 | 0.2063 | 0.1953 | 0.3381 | −0.2229 | 0.0577 | 0.1992 | 0.4241 | 0.2030 | −0.1337 | 0.2088 |
| 39 | −0.3112 | −0.2523 | −0.2838 | −0.3482 | 0.1553 | −0.0911 | 0.0574 | −0.1409 | 0.4294 | 0.3296 | −0.0641 | −0.1471 | −0.3774 | −0.1647 | −0.1976 | 0.2069 | −0.1889 | −0.0439 | 0.2317 | −0.0299 |

**Table 19.** *Cont.*

| Neuron | 21 | 22 | 23 | 24 | 25 | 26 | 27 | 28 | 29 | First Hidden Layer 30 | 31 | 32 | 33 | 34 | 35 | 36 | 37 | 38 | 39 | 40 | Bias |
|---|---|---|---|---|---|---|---|---|---|---|---|---|---|---|---|---|---|---|---|---|---|
| 1 | 0.2226 | −0.2608 | 0.0453 | 0.1842 | 0.3802 | 0.3290 | 0.1709 | −0.0622 | 0.0024 | 0.0261 | −0.1708 | −0.3243 | 0.3717 | −0.2252 | −0.1437 | −0.1778 | −0.1762 | −0.0297 | 0.2895 | 0.0446 | −1.5575 |
| 2 | 0.2719 | −0.2400 | −0.1186 | 0.3518 | 0.2427 | 0.2026 | 0.3053 | −0.4083 | 0.3909 | −0.0256 | −0.1569 | −0.2815 | −0.0096 | −0.1602 | 0.2972 | −0.2061 | 0.3052 | −0.3843 | 0.0607 | 0.1708 | −1.4321 |
| 3 | 0.1953 | 0.0773 | −0.4534 | 0.3047 | 0.0785 | −0.4133 | −0.2228 | 0.3620 | 0.1284 | −0.2609 | 0.1174 | 0.4776 | −0.2847 | 0.2270 | 0.2265 | −0.4136 | 0.2257 | 0.4271 | 0.0681 | 0.0936 | −1.3873 |
| 4 | −0.0985 | 0.3888 | 0.2197 | 0.1470 | 0.4458 | −0.3077 | −0.1670 | 0.0169 | −0.0204 | 0.1570 | −0.3843 | 0.2875 | 0.3486 | 0.1509 | 0.2176 | −0.2237 | −0.0117 | 0.1510 | −0.0486 | 0.3012 | −1.2745 |
| 5 | −0.3232 | 0.3340 | −0.3841 | 0.1154 | 0.3722 | −0.2058 | −0.0271 | −0.3964 | −0.1351 | −0.1587 | −0.0116 | −0.2605 | −0.0021 | 0.1012 | −0.0967 | −0.2644 | −0.2715 | −0.1702 | 0.3740 | 0.2941 | 1.2221 |
| 6 | −0.0349 | −0.1820 | 0.0064 | 0.4890 | −0.3517 | −0.0457 | 0.1152 | 0.3051 | 0.3575 | −0.4647 | 0.0218 | 0.1592 | 0.1771 | −0.2999 | −0.0445 | 0.2098 | −0.3868 | −0.1852 | −0.0070 | 0.2247 | −1.1044 |
| 7 | −0.1340 | −0.3960 | 0.3151 | 0.0273 | 0.1221 | −0.3383 | −0.1558 | 0.3246 | −0.1259 | −0.0895 | −0.0662 | 0.2196 | 0.1996 | 0.2432 | 0.3399 | 0.4207 | −0.1968 | −0.1638 | −0.1626 | 0.2776 | 1.0731 |
| 8 | −0.1941 | −0.2767 | −0.1628 | −0.0663 | −0.1757 | −0.2890 | 0.3802 | 0.1999 | 0.2850 | −0.1252 | 0.3092 | 0.0691 | 0.1730 | 0.2758 | −0.3105 | −0.1401 | −0.2917 | 0.0089 | −0.3895 | 0.2400 | 0.9720 |
| 9 | −0.3102 | 0.1905 | 0.0054 | 0.4023 | 0.1720 | 0.2708 | 0.2719 | −0.1633 | 0.3319 | 0.3041 | 0.0221 | 0.0064 | 0.0560 | 0.3137 | 0.1141 | 0.4263 | 0.2396 | −0.2858 | 0.0791 | −0.4544 | −0.8734 |
| 10 | 0.1883 | −0.3968 | 0.3798 | −0.1657 | −0.3879 | 0.2453 | −0.2370 | 0.1195 | 0.2504 | 0.2241 | 0.2052 | −0.0853 | 0.2344 | 0.3072 | 0.1086 | 0.0918 | 0.2928 | −0.0868 | 0.4908 | 0.3090 | 0.8089 |
| 11 | −0.1266 | −0.1461 | −0.0775 | −0.1708 | −0.2759 | −0.1844 | 0.2305 | 0.2191 | −0.3120 | 0.2915 | 0.3444 | 0.2912 | 0.3031 | −0.3567 | −0.3334 | 0.2673 | 0.0425 | 0.2097 | −0.3751 | −0.3990 | −0.7268 |
| 12 | −0.2875 | −0.4759 | −0.2336 | −0.1944 | −0.0694 | −0.0782 | −0.3519 | 0.2234 | 0.3426 | −0.4138 | 0.2385 | −0.2739 | −0.2417 | 0.0311 | −0.1176 | 0.3008 | −0.1868 | 0.0218 | 0.2220 | −0.0207 | 0.6471 |
| 13 | 0.0945 | −0.0462 | 0.1885 | −0.3065 | −0.1065 | −0.1757 | 0.3354 | −0.2363 | 0.0205 | −0.2940 | 0.2696 | 0.0497 | 0.3650 | −0.1506 | 0.2480 | 0.2689 | −0.1105 | 0.0953 | −0.2462 | 0.1376 | 0.5823 |
| 14 | −0.3362 | 0.3231 | −0.1162 | −0.1109 | −0.3146 | −0.3602 | 0.1314 | −0.6308 | 0.1263 | −0.0394 | −0.0227 | 0.0407 | −0.3351 | −0.1949 | −0.4061 | 0.1358 | −0.2797 | −0.1068 | 0.2548 | 0.1865 | −0.4488 |
| 15 | −0.1203 | 0.1658 | 0.3925 | −0.0093 | 0.0588 | −0.3781 | −0.2402 | 0.2309 | 0.1481 | −0.2758 | 0.1338 | −0.3087 | −0.3797 | 0.2061 | 0.2861 | −0.0862 | −0.1171 | 0.2761 | 0.2922 | 0.3641 | 0.4241 |
| 16 | −0.1530 | 0.1709 | −0.1940 | −0.0764 | −0.2111 | 0.1085 | 0.1984 | 0.2636 | −0.4966 | 0.2216 | 0.2766 | 0.5385 | 0.3877 | −0.3252 | 0.1792 | 0.1171 | −0.2994 | 0.2245 | −0.1811 | −0.2816 | 0.3017 |
| 17 | −0.0077 | −0.1449 | 0.1344 | −0.0591 | 0.0105 | 0.0791 | 0.3838 | −0.2934 | −0.2890 | −0.1440 | −0.4468 | −0.3551 | −0.4352 | −0.2409 | 0.0351 | −0.2100 | 0.0776 | 0.0714 | −0.1592 | −0.0980 | 0.2468 |
| 18 | 0.2247 | −0.2536 | −0.0402 | 0.5374 | −0.2511 | 0.3167 | 0.4253 | −0.1168 | 0.3176 | −0.1549 | −0.2237 | 0.0989 | −0.2494 | −0.2615 | −0.2052 | −0.1912 | 0.4256 | −0.0761 | 0.0875 | −0.2739 | −0.1677 |
| 19 | −0.3651 | 0.1756 | 0.1493 | −0.2964 | 0.2036 | 0.1695 | −0.0040 | −0.2456 | −0.3901 | 0.1820 | −0.1683 | −0.3968 | 0.1064 | −0.0707 | −0.1593 | −0.4150 | 0.1546 | −0.2778 | −0.4154 | 0.0397 | 0.0908 |
| 20 | 0.2865 | 0.3705 | 0.3068 | 0.1852 | −0.2399 | −0.1673 | −0.0507 | 0.2960 | 0.0209 | 0.3317 | −0.3333 | −0.2280 | 0.2478 | −0.1563 | −0.2799 | −0.1685 | −0.1899 | 0.1223 | −0.2100 | 0.2353 | −0.0041 |
| 21 | 0.3414 | −0.2998 | 0.2059 | −0.2379 | −0.1476 | −0.2143 | 0.2478 | −0.2011 | −0.1115 | 0.4529 | −0.3283 | 0.2279 | 0.4156 | −0.3613 | 0.2046 | 0.2386 | 0.3757 | 0.0968 | −0.3174 | 0.0175 | 0.0343 |
| 22 | −0.1978 | −0.1063 | 0.0800 | 0.4500 | 0.1092 | 0.0955 | 0.3156 | −0.3378 | 0.0930 | −0.2154 | −0.2246 | −0.1173 | 0.2206 | −0.1957 | −0.0317 | 0.0603 | 0.3201 | −0.0577 | 0.2594 | 0.2214 | 0.1757 |
| 23 | −0.2409 | 0.1009 | 0.2247 | −0.1998 | −0.0660 | −0.4355 | 0.1241 | −0.0855 | −0.3083 | −0.1067 | −0.4979 | −0.2220 | −0.1595 | 0.2971 | 0.2765 | 0.1363 | 0.0806 | 0.3962 | −0.1198 | −0.1371 | 0.2539 |
| 24 | −0.4017 | 0.0301 | 0.3170 | 0.2072 | −0.0273 | −0.1784 | 0.1676 | −0.3540 | 0.2144 | 0.3080 | 0.1259 | 0.1311 | −0.0598 | 0.2239 | −0.2608 | −0.3756 | −0.0302 | 0.3005 | 0.3345 | 0.1981 | 0.3645 |
| 25 | −0.0359 | −0.1898 | 0.2651 | 0.0568 | −0.1194 | −0.2595 | −0.3507 | 0.1059 | −0.2591 | −0.2291 | 0.3213 | −0.2817 | 0.1568 | 0.3176 | −0.0997 | −0.3055 | 0.2819 | −0.4165 | 0.1084 | 0.2803 | −0.4166 |
| 26 | 0.1062 | −0.3647 | −0.1803 | 0.0608 | 0.3979 | −0.0412 | −0.1382 | 0.3910 | 0.3869 | 0.0477 | 0.1759 | −0.0841 | −0.0487 | −0.1390 | −0.3939 | −0.4250 | −0.2347 | −0.0489 | 0.0914 | −0.1004 | 0.4910 |
| 27 | 0.3074 | −0.1479 | −0.4109 | 0.1461 | −0.3724 | −0.4238 | 0.0204 | 0.0427 | −0.0552 | −0.1938 | −0.0776 | −0.3518 | 0.3355 | 0.2503 | 0.3114 | 0.1863 | 0.2145 | 0.0846 | −0.3659 | −0.1335 | −0.5483 |
| 28 | −0.1918 | −0.1527 | −0.1167 | −0.0149 | −0.0741 | −0.2527 | 0.3753 | 0.1018 | −0.3310 | 0.0413 | −0.1289 | 0.2394 | 0.0917 | 0.3111 | 0.2693 | −0.2638 | 0.3091 | −0.0517 | −0.0540 | −0.3865 | 0.6372 |
| 29 | −0.2643 | −0.2059 | 0.0341 | −0.1394 | −0.3118 | 0.3653 | −0.2946 | −0.2437 | −0.0694 | −0.1189 | −0.0010 | 0.4408 | 0.3802 | 0.1184 | 0.3108 | −0.4254 | −0.0029 | 0.2538 | 0.3449 | 0.0219 | −0.6989 |
| 30 | 0.4718 | −0.2049 | −0.0080 | 0.0787 | −0.4437 | 0.3237 | −0.2115 | −0.4133 | 0.0253 | 0.0864 | 0.1934 | −0.0010 | −0.0245 | 0.1939 | 0.3577 | 0.3385 | −0.0846 | −0.2007 | −0.2413 | 0.1990 | −0.8035 |
| 31 | −0.0856 | −0.1866 | 0.1180 | −0.0087 | −0.2604 | −0.2210 | 0.3461 | −0.4021 | −0.0308 | −0.3325 | 0.1662 | 0.0925 | −0.4058 | −0.1056 | 0.1260 | 0.0943 | −0.2518 | −0.3059 | −0.2800 | −0.2081 | 0.8609 |
| 32 | −0.3001 | −0.1203 | 0.0977 | 0.1225 | −0.1566 | −0.1078 | 0.0368 | 0.1114 | −0.1031 | −0.0498 | −0.1805 | 0.3014 | 0.2247 | −0.2616 | −0.3101 | −0.4321 | 0.3029 | 0.0782 | 0.3135 | 0.3556 | −0.9600 |
| 33 | 0.2239 | −0.3574 | −0.3657 | −0.1378 | −0.1040 | −0.1216 | −0.2233 | −0.4116 | 0.0662 | −0.0371 | −0.4825 | 0.4074 | 0.1529 | 0.2037 | −0.0707 | −0.0494 | 0.2155 | 0.2578 | −0.2437 | 0.1757 | 1.0187 |
| 34 | 0.0368 | 0.2221 | −0.3119 | −0.3275 | 0.1182 | 0.3658 | −0.1383 | −0.0264 | −0.4255 | 0.4009 | −0.2236 | −0.3385 | 0.4065 | −0.0556 | −0.0839 | −0.0818 | −0.3409 | 0.2607 | −0.1398 | 0.1946 | 1.0709 |
| 35 | −0.1744 | −0.0638 | 0.2230 | −0.1008 | 0.0888 | −0.3382 | −0.2316 | 0.2853 | −0.2672 | −0.2439 | 0.1251 | 0.2545 | 0.2314 | 0.1470 | 0.1274 | −0.3669 | 0.2321 | −0.2944 | −0.2316 | −0.3928 | 1.1753 |
| 36 | 0.1350 | 0.2189 | −0.1752 | −0.0635 | 0.2924 | 0.3223 | −0.2680 | 0.0014 | 0.2019 | −0.0811 | −0.2209 | −0.2380 | −0.2323 | −0.3617 | −0.0488 | −0.3692 | −0.3753 | −0.3377 | 0.0469 | −0.2969 | −1.3156 |
| 37 | 0.2099 | 0.1271 | 0.0223 | −0.2607 | 0.3721 | 0.3413 | 0.1485 | −0.2894 | −0.0429 | 0.1758 | −0.4344 | −0.0366 | −0.3691 | 0.3444 | 0.4241 | 0.2995 | 0.0680 | −0.1413 | 0.1336 | 0.1817 | −1.3629 |
| 38 | 0.3851 | −0.2410 | −0.0101 | −0.1786 | 0.3164 | 0.3778 | 0.1264 | 0.2387 | 0.2891 | −0.2700 | 0.0743 | 0.1923 | 0.2754 | −0.0541 | 0.2902 | −0.1209 | −0.2282 | 0.0630 | −0.0635 | −0.3089 | 1.4434 |
| 39 | −0.0616 | 0.3390 | 0.4548 | 0.1988 | 0.3229 | 0.2196 | 0.1233 | 0.4229 | −0.2809 | 0.2354 | −0.1973 | −0.1215 | 0.2145 | −0.1600 | 0.0229 | 0.2061 | −0.1263 | −0.1236 | −0.2105 | −0.4252 | −1.5342 |

**Table 20.** Neurons, weights, and biases for the output layer for the simulated ANN model.

| | 1 | 2 | 3 | 4 | 5 | 6 | 7 | 8 | 9 | Second Hidden Layer Neuron 10 | 11 | 12 | 13 | 14 | 15 | 16 | 17 | 18 | 19 | 20 |
|---|---|---|---|---|---|---|---|---|---|---|---|---|---|---|---|---|---|---|---|---|
| Weight | 0.8320 | −0.3520 | −0.6236 | 0.8300 | 0.2183 | 0.4836 | 0.3338 | 1.0340 | −0.4871 | −0.9513 | −0.5188 | −0.2229 | 0.5155 | −0.7244 | 0.1526 | −0.6513 | 0.0370 | −0.3207 | −0.3903 | −0.3953 |

| | 21 | 22 | 23 | 24 | 25 | 26 | 27 | 28 | 29 | Second Hidden Layer Neuron 30 | 31 | 32 | 33 | 34 | 35 | 36 | 37 | 38 | 39 | Bias |
|---|---|---|---|---|---|---|---|---|---|---|---|---|---|---|---|---|---|---|---|---|
| Weight | −0.7215 | −0.7665 | −0.7620 | 0.2172 | 0.2657 | 0.8019 | −0.3311 | −0.7170 | −0.3379 | 0.1952 | −0.4010 | 0.4523 | 0.8320 | −0.7341 | −0.2346 | 0.4236 | 0.2719 | −0.7039 | 0.0767 | −0.7918 |

Based on the overall correlation coefficient and mean squared errors of the ANN model, as mentioned above, the model can provide a good estimation of the dissociation constants at various temperatures. When comparing the full, reduced, and simulated ANN models with the same ANN architecture, the overall correlation coefficients ($R_{overal}$), and mean squared errors (MSEs) for overall, training, validating, and testing of the three models were not very different. The simulated model has even lower mean squared errors with higher overall correlation coefficients than the other models. In addition, the simulated model is more convenient than the others because it uses the simulated data for the estimations.

## 9. Conclusions

The first and second p$K_a$ of 3-(Diethylamino) propylamine, 1,3-Diaminopentane, 3-Butoxypropylamine, 2-(Methylamino) ethanol, Bis(2-methoxyethyl) amine, α-Methylbenzylamine, 2-Aminoheptane, and 3-Amino-1-phenylbutane were experimentally measured from the temperature range between 293.15 K to 313.15 K with 5 K increments and 323.15 K at atmospheric pressure. The dissociation constants of the amines decreased when the temperature increased. In addition, by employing the Van't Hoff equation, the thermodynamic quantities of the reactions were determined for the eight amines studied. Among the amines, 3-(Diethylamino) propylamine and 1,3-Diaminopentane have unsymmetrical structures; it is, therefore, necessary to determine the protonation reaction orders for these amines. Computational chemistry calculations confirmed that the first p$K_a$ values of 3-(Diethylamino) propylamine were associated with the tertiary amino while the first p$K_a$ values of 1,3-Diaminopentane were associated with the primary amino group on the carbon chain.

In addition, the dissociation constant values of the studied amines at 298.15 K were calculated using functional group and computational chemistry methods. The paper–pencil methods provided faster calculations while keeping the predicted p$K_a$ values at acceptable accuracy. As the paper–pencil methods could not determine the dissociation constants at any temperature but at 298 K, computational chemistry methods were used to perform the calculations using two different thermodynamic cycles. The two cycles resulted in different predictions. However, the calculated values were not very accurate; therefore, artificial neural network (ANN) models were tested to provide better accuracy. The models simply collected and combined experimental data of many amines to estimate the p$K_a$. Although the ANN models can produce very good estimations, the main drawback was to obtain the experimental data. If the required data were not available or the chemicals had not been synthesized, the dissociation constant values could not be determined. Instead of using the experimental properties, the software can be used to generate the data for the ANN model. The simulated ANN model was able to fit the experimental p$K_a$ quite well.

**Supplementary Materials:** The following supporting information can be downloaded at: https://www.mdpi.com/article/10.3390/liquids3020016/s1. References [26–64] are cited in the supplementary materials.

**Author Contributions:** Conceptualization, A.H. and W.N.; methodology, V.S.P.V.A. and W.N.; software, W.N. and V.S.P.V.A.; validation, V.S.P.V.A. and W.N.; formal analysis, V.S.P.V.A. and W.N.; investigation, V.S.P.V.A. and W.N.; resources, A.H.; data curation, V.S.P.V.A. and W.N.; writing—original draft preparation, V.S.P.V.A. and W.N.; writing—review and editing, A.H.; visualization, W.N.; supervision, A.H.; project administration, A.H. and W.N.; funding acquisition, A.H. All authors have read and agreed to the published version of the manuscript.

**Funding:** This work was funded by the Natural Science and Engineering Council (NSERC), Canada, in the form of a Discovery Grant.

**Data Availability Statement:** Data used in this manuscript can be found in the Supplementary Materials Section.

**Acknowledgments:** The authors wish to express their appreciation to the Natural Sciences and Engineering Research Council (Canada) for providing a Discovery Grant to the corresponding author, and an NSERC PGS D scholarship to the first author.

**Conflicts of Interest:** The authors declare no competing financial interests.

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
