# Peer review of "Determination of the Dissociation Constants (pKa) of Eight Amines of Importance in Carbon Capture: Computational Chemistry Calculations, and Artificial Neural Network Models"

_liquids, doi:10.3390/liquids3020016_

Round 1
Reviewer 1 Report
This work focuses on determining the dissociation constants of eight amines within temperatures ranging from 293.15 K to 323.15 K. The thermodynamic properties of the protonated reactions were regressed from the pKa work. In addition, the protonated order of both 3-(Diethylamino) propylamine and 1, 3-Diaminopentane were determined using computational chemistry methods owing to their unsymmetrical structures. Besides the experimental methods, the dissociation constants at the standard temperature were also estimated using group functional models (paper-pencil) and computational methods. This work is interesting and thus can be accepted after addressing the following issues.
1. The raw data is too much and can be considered to be put into supporting information. Some of the data in the tables can be considered to be presented in the form of graphs.
2. The results of the 8 amine solvents should be compared to the MEA solvent results.
3. Is this proposed ANN model of general utility? For example, is it applicable to blended amine solvents?
No
Reviewer 2 Report
This work focuses on determining the dissociation constant of some amines. The study is quite extensive and shows the effort of the researchers. However, the manuscript is not well structured and becomes difficult to read.
Here are some suggestions for improving the manuscript:
1.- Add the Methods section, where the document's outline is schematized, and how the results are organized in the manuscript.
2.- Do not repeat the same information or data in tables and figures (examples: Table 2 and Figure 1 show the same information, Table 3 and Figure 2 idem).
3.- In Figure 2, please explain what superscripts mean (VI, V2, VI, V2).
4. Section 8 would be restructured. Reporting the pKa data and its comparisons with experimental or reference data is suggested. See the document (doi: 10.2166/wst.2022.186) for consideration in the report of studies based on ANNs. In addition, delete Tables 13-15 and 17-21 or send them to Supporting Information.
5. Review the number of significant figures reported in the figures and tables, considering 3-4 decimals.
Minor suggestions:
There are different types and sizes of letters in some sections. Please unify.
Moderate editing of English language.
Round 2
Reviewer 1 Report
The MS can be accepted.
Reviewer 2 Report
In general, I believe that the authors have not responded convincingly to the questions I asked during my first review of the paper. I think the study is quite extensive and shows the effort of the researchers. However, the manuscript is not well structured and becomes difficult to read.
n/a